# Double Gumbel Q-Learning

**David Yu-Tung Hui**
Mila, Université de Montréal
`dythui2+drl@gmail.com`

**Aaron Courville**
Mila, Université de Montréal

**Pierre-Luc Bacon**
Mila, Université de Montréal

## Abstract

We show that Deep Neural Networks introduce two heteroscedastic Gumbel noise sources into Q-Learning. To account for these noise sources, we propose Double Gumbel Q-Learning, a Deep Q-Learning algorithm applicable for both discrete and continuous control. In discrete control, we derive a closed-form expression for the loss function of our algorithm. In continuous control, this loss function is intractable and we therefore derive an approximation with a hyperparameter whose value regulates pessimism in Q-Learning. We present a default value for our pessimism hyperparameter that enables DoubleGum to outperform DDPG, TD3, SAC, XQL, quantile regression, and Mixture-of-Gaussian Critics in aggregate over 33 tasks from DeepMind Control, MuJoCo, MetaWorld, and Box2D and show that tuning this hyperparameter may further improve sample efficiency.

## 1 Introduction

Reinforcement Learning (RL) algorithms learn optimally rewarding behaviors (Sutton and Barto, 1998, 2018). Recent RL algorithms have attained superhuman performance in Atari (Mnih et al., 2015), Go (Silver et al., 2016), Chess (Schrittwieser et al., 2020), StarCraft (Vinyals et al., 2019), simulation racing (Wurman et al., 2022), and have also been used to control stratospheric balloons (Bellemare et al., 2020) and magnetic fields in Tokamak nuclear fusion reactors (Degrave et al., 2022). All these aforementioned agents use a $Q$-function (Watkins and Dayan, 1992) to measure the respective quality of their behaviors.

Many algorithms (Lillicrap et al., 2015; Mnih et al., 2016, 2015; Schulman et al., 2015; Fujimoto et al., 2018; Haarnoja et al., 2018b) learn a $Q$-function by minimizing the Mean-Squared Bellman Error (MSBE). The functional form of the MSBE was motivated by an analogy to the popular Mean-Squared Error (MSE) in supervised learning regression problems, but there is little theoretical justification for its use in RL (Bradtke and Barto, 1996; Ernst et al., 2005; Riedmiller, 2005).

Training a deep neural network to minimize the MSBE is empirically unstable (Irpan, 2018; Henderson et al., 2018; Van Hasselt et al., 2016) because deep neural networks induce overestimation bias in the $Q$-function (Thrun and Schwartz, 1993; Fujimoto et al., 2018). A popular method to reduce overestimation 'pessimistically' selects less positive $Q$-values from an ensemble of $Q$-functions, effectively returning low-quantile estimates (Fujimoto et al., 2018; Kuznetsov et al., 2020; Ball et al., 2023). However, an ensemble is computationally expensive and quantiles only provide discrete-grained control over the degree of pessimism. Curiously, pessimism has mainly been applied to 'continuous control' problems that require continuous-valued actions and not discrete control.

This paper analyzes noise in Deep $Q$-Learning. We introduce a loss function for Deep $Q$-Learning and a scalar hyperparameter that adjusts pessimism without an ensemble. These two innovations form **Double Gumbel $Q$-Learning (DoubleGum)**, [1] an off-policy Deep $Q$-Learning algorithm applicable to both discrete and continuous control. Our paper is structured as follows:

1. Section 2.1 shows that the MSBE implicitly uses a noise model of the Bellman Optimality Equation with additive homoscedastic noise. Section 4.1 shows that this noise model is empirically too coarse to fully capture the distribution of noise in Deep $Q$-Learning.

2. Section 3 derives a noise model of the Soft Bellman Equation with additive heteroscedastic logistic noise derived from two heteroscedastic Gumbel noise sources. Section 4.1 shows that our noise model is a better empirical fit to the noise distribution in Deep $Q$-Learning.

3. Section 3.3 shows that when the soft value function in the Soft Bellman Equation is intractable in continuous control, its approximation leads to methods that adjust pessimism. Section 3.1 presents a method to adjust pessimism with a single scalar hyperparameter, and its effect is empirically verified in Section 4.2.

4. Section 5 discusses related work, notably presenting DoubleGum as a special case of Distributional RL and a generalization of existing Maximum Entropy (MaxEnt) RL algorithms.

5. Section 6 benchmarks DoubleGum, showing marginal improvements over DQN baselines on 3 classic discrete control tasks and improved aggregate performance over DDPG, SAC, TD3, XQL and two Distributional RL algorithms over 33 continuous control tasks.

## 2   Mathematical Background

Reinforcement Learning agents learn to make maximally rewarding decisions in an environment (Sutton and Barto, 1998, 2018). In Markov Decision Process (MDP) environments, a $Q$-function specifies the expected amount of reward an agent obtains (Watkins and Dayan, 1992) and is also known as a state-action value function. MDPs define a transition probability $p(s' \mid s, a)$ and reward $r(s, a, s')$ for every pair of states $s, s' \in \mathcal{S}$ and action $a \in \mathcal{A}$ (Bellman, 1957; Howard, 1960). Decisions are made by a policy $\pi(a \mid s)$ at every discrete timestep and the $Q$-function of $\pi$ at $s, a$ is

$$Q_\pi(s, a) = \mathbb{E}_{\pi, p} \left[ \sum_{j=i}^{n} \gamma^j r_j \middle| s_i = s, a_i = s \right], \text{ denoting } r(s_j, a_j, s_{j+1}) \text{ as } r_j,$$

where the discount factor $\gamma \in ]0, 1[$ ensures finite $Q$ for infinite time-horizon $n$ (Samuelson, 1937). The expectation in the $Q$-function is computed over a trajectory sequence of $s$ and $a$ induced by the coupling of the policy and the transition function.

In every MDP, an optimal policy $\pi^\star$ makes maximally rewarding decisions following $s \to \arg\max_a Q^\star(s, a)$ (Watkins and Dayan, 1992). $Q^\star$ is the optimal value function specified as the unique fixed point of the Bellman Optimality Equation of the MDP (Bellman, 1957)

$$Q^\star(s, a) = \mathbb{E}_{p(s'|s,a)} \left[ r + \gamma \max_{a'} Q^\star(s', a') \right], \text{ over all } s \text{ and } a, \text{ denoting } r(s, a, s') \text{ as } r. \quad (1)$$

In Deep $Q$-Learning, $Q^\star$ is approximated by a neural network $Q_\theta$ with parameters $\theta$. Following Munos and Szepesvári (2008), $\theta$ is updated by a sequence of optimization problems whose $i^{\text{th}}$ stage iteration is given by

$$\theta_{i+1} = \arg\min_\theta \mathrm{L}(Q_\theta, y_{\bar{\theta}_i}), \text{ where } y_i(s, a) = \mathbb{E}_{p(s'|s,a)} \left[ r + \gamma \max_{a'} Q_{\bar{\theta}_i}(s', a') \right],$$

where the 'stop-gradient' operator $\overline{(\cdot)}$ denotes evaluating but not optimizing an objective with respect to its argument and $y$ is dubbed the bootstrapped target. In this work, we restrict our environments to deterministic MDPs, so the resultant expectation over $s'$ is computed over one sample.

Typically, the loss function $\mathrm{L}(Q_\theta, y_{\bar{\theta}_i}) = \mathbb{E}_{s,a} \left( Q_\theta(s, a) - y_{\bar{\theta}_i}(s, a) \right)^2$ is the Mean-Squared Bellman Error (MSBE). At every $i$, L is minimized with stochastic gradient descent by a fixed number of gradient descent steps and not optimized to convergence (Mnih et al., 2015; Lillicrap et al., 2015).

---

[1]Code: https://github.com/dyth/doublegum

## 2.1 Deriving the Mean-Squared Bellman Error

Thrun and Schwartz (1993) model $Q_\theta$(s, a) as a random variable whose value varies with $\theta$, $s$ and $a$. They set up a generative process $Q_\theta(s,a) = Q^\star(s,a) - \epsilon_{\theta,s,a}$ showing that the function approximator $Q_\theta$ is a noisy function of the implicit true value $Q^\star$, but they do not make any assumptions about $\epsilon$. [2]

We derive the MSBE with two additional assumptions: $\epsilon_{\theta,s,a} \overset{\text{iid}}{\sim} \mathcal{N}(0, \sigma^2)$, where $\mathcal{N}$ is a homoscedastic normal due to the Central Limit Theorem (CLT) [3] and secondly $Q^\star(s,a) \approx y_{\bar{\theta}}(s,a)$, where $\theta$ are the parameters at an arbitrary optimization stage. Incorporating these two assumptions into the noise model yields $y_{\bar{\theta}}(s,a) = Q_\theta(s,a) + \epsilon_{\theta,s,a}$, uncovering the implicit assumption that Temporal-Difference (TD) errors $y_{\bar{\theta}} - Q_\theta$ follow a homoscedastic normal. Maximum Likelihood Estimation (MLE) of $\theta$ on the resultant noise model yields the MSBE after abstracting away the constants

$$\max_\theta \mathbb{E}_{s,a} \log p(y_{\bar{\theta}}(s,a)) = \min_\theta \mathbb{E}_{s,a} \left[ \log \sigma\sqrt{2\pi} + \frac{1}{\sigma^2} \left( Q_\theta(s,a) - y_{\bar{\theta}}(s,a) \right)^2 \right] \ .$$

## 2.2 The Limiting Distribution in Bootstrapped Targets

Both assumptions used to derive the MSBE are theoretically weak. First, the CLT states that iid samples of an underlying distribution tend towards a homoscedastic normal but do not account for the form of the underlying distribution. Secondly, there is no analytic justification for $Q^\star(s,a) \approx y_{\bar{\theta}}(s,a)$. Substituting the Thrun and Schwartz (1993) model in the RHS of the Bellman Optimality Equation yields

$$Q^\star(s,a) = \mathbb{E}_{p(s'|s,a)} \left[ r + \gamma \max_{a'}[Q_\theta(s',a') + \epsilon_{\theta,s',a'}] \right] = \mathbb{E}_{p(s'|s,a)} [r + \gamma g_{\theta,s'}] \neq y_{\bar{\theta}}(s,a) \ .$$

When the $\max$ over a finite number of iid samples with unbounded support is taken, the Extreme Value Theorem (EVT) gives the limiting distribution of the resultant random variable as a Gumbel distribution (Fisher and Tippett, 1928; Gnedenko, 1943), defined here as $g_{\theta,s'} \sim \mathcal{G}(\alpha, \beta)$. Expressions relating $\alpha$ and $\beta$ to the parameters of the underlying iid random variables are rarely analytically expressible (Kimball, 1946; Jowitt, 1979) and there is no guarantee $Q^\star(s,a) \approx y_{\bar{\theta}}(s,a)$.

# 3 Deriving a Deep Q-Learning Algorithm from First Principles

We find an analytic expression that relates the limiting Gumbel distribution to parameters of the underlying noise distribution from function approximation. To do so, we assume that noise in the Thrun and Schwartz (1993) model is a heteroscedastic Gumbel noise with state-dependent spread

$$Q_\theta(s,a) = Q^\star(s,a) - g_{\theta,a}(s), \quad g_{\theta,a}(\cdot) \overset{\text{iid}}{\sim} \mathcal{G}(0, \beta_\theta(\cdot)), \ \text{ for all } a \in \mathcal{A} \ , \tag{2}$$

Appendix A.1 restates the finding of McFadden et al. (1973) and Rust (1994), which results in the following expression with the soft value function $V_\theta^{\text{soft}}$

$$\max_a Q^\star(s,a) = \max_a \left[ Q_\theta(s,a) + g_{\theta,a}(s) \right] = \beta_\theta(s) \log \int \exp\left( \frac{Q_\theta(s,a)}{\beta_\theta(s)} \right) \, \mathrm{d}a + g_\theta(s)$$

$$= V_\theta^{\text{soft}}(s) + g_\theta(s), \ \text{ where } g_{\theta,a}(\cdot), g_\theta(\cdot) \overset{\text{iid}}{\sim} \mathcal{G}(0, \beta_\theta(\cdot)) \ . \tag{3}$$

Substituting Equations 2 and 3 into 1 yields a new noise model of the Soft Bellman Optimality Equation with two function approximators and two heteroscedastic Gumbel noise sources

$$Q_\theta(s,a) + g_{\theta,a}(s) = \mathbb{E}_{p(s'|s,a)} \left[ r + \gamma V_{\bar{\theta}}^{\text{soft}}(s') + \gamma g_{\bar{\theta}}(s') \right] \ . \tag{4}$$

We now develop this noise model into a loss function used in our algorithm, DoubleGum.

---

[2]This equation derives from an unnumbered equation at the top of Page 3 of Thrun and Schwartz (1993), and we have renamed their variables to fit our notation. The authors add the noise $\epsilon$ to $Q^\star$, but we subtract it for notational clarity later on without loss of generality. We additionally subscript the noise by $\theta$ to clarify that the random variable is resampled for different $\theta$s in the function approximator.

[3]Ernst et al. (2005) and Riedmiller (2005) introduced the MSBE to $Q$-Learning, arguing that MSE was popular for regression problems in supervised learning. In regression, the MSE arises from MLE parameter estimation of a homoscedastic normal, which is assumed to exist because of the CLT.

## 3.1 Merging Two Distributions

We replace $Q_\theta$ with a new function approximator $Q_\theta^{\text{new}}(s, a) = Q_\theta(s, a) + g_\theta(s)$, resulting in a new soft value function on the RHS of Equation 4

$$V_\theta^{\text{soft}}(s) + g_\theta(s) = \beta_\theta(s) \log \int \exp\left(\frac{Q_\theta^{\text{new}}(s, a)}{\beta_\theta(s)}\right) \, \mathrm{d}a = V_\theta^{\text{soft, new}}(s) \ \text{ (Appendix A.2)},$$

and a LHS with a heteroscedastic logistic random variable obtained in Appendix A.3 from the difference of two Gumbel random variables with equation spreads. Note that this operation is only permissible in MDPs without terminating states. Equation 4 simplifies into

$$Q_\theta^{\text{new}}(s, a) + l_{\theta, a}(s) = \underset{p(s'|s,a)}{\mathbb{E}}\left[r + \gamma V_{\bar\theta}^{\text{soft, new}}(s')\right], \ \ l_{\theta, a}(\cdot) \sim \mathcal{L}\left(0, \beta_\theta(\cdot)\right) \ \ , \tag{5}$$

the Soft Bellman Equation with heteroscedastic logistic noise. The bootstrapped targets in this equation is the RHS, which we will denote by $y_{\bar\theta}^{\text{soft}}(s, a)$.

## 3.2 Parameter Estimation with Moment Matching

Equation 5 does not allow MLE of $\theta$ because the heteroscedastic logistic is not a member of the exponential family and lacks a sufficient statistical estimator. We learn $\theta$ with moment matching by MLE of a heteroscedastic normal $y_{\bar\theta}^{\text{soft}}(s, a) \sim \mathcal{N}(Q_\theta^{\text{new}}(s, a), \sigma_\theta(s))$ given by

$$\min_\theta \left[\log \sigma_\theta(s) + \left(\frac{y_{\bar\theta}^{\text{soft}}(s, a) - Q_\theta^{\text{new}}(s, a)}{\sigma_\theta(s)}\right)^2\right] \ \ , \tag{6}$$

and recovering the logistic location and spread as $Q_\theta^{\text{new}}(s, a)$ and $\beta_\theta(s) = \sigma_\theta(s)\frac{\sqrt{3}}{\pi}$.

## 3.3 Computing the Bootstrapped Targets

$y_\theta^{\text{soft}}$ is calculated with one sample from $s'$ and $a'$ where appropriate. In discrete control, $\mathcal{A}$ is finite and the expression $V_\theta^{\text{soft, new}}$ is a sum. The resultant bootstrapped target in discrete control is

$$y_\theta^{\text{discrete}}(s, a, r, s') = r + \gamma V_\theta^{\text{soft, new}}(s') \ \ .$$

In continuous control, $V_\theta^{\text{soft, new}}$ is intractable because $\mathcal{A}$ contains infinitesimally small quantities. Appendix A.4, adapted from Equation 19 in Haarnoja et al. (2017), variationally approximates the integral with respect to a policy neural network $\pi_\phi(a \mid s)$ with parameters $\phi$, producing

$$V_\theta^{\text{soft, new}}(s) = \underset{\pi_\phi(a|s)}{\mathbb{E}}[Q_\theta^{\text{new}}(s, a)] + \beta_\theta(s) \, \mathbb{C}[\pi_\phi \mid\mid p_\theta], \ \text{ where } p_\theta(a \mid s) \propto \exp\frac{Q_\theta^{\text{new}}(s, a)}{\beta_\theta(s)} \ \ . \tag{7}$$

Denoting $\mathbb{C}[\pi_\phi \mid\mid p_\theta]$ as a constant $\frac{\pi}{\sqrt{3}}c$ yields bootstrapped targets of

$$y_\theta^{\text{continuous}}(s, a, r, s') = r + \gamma \underset{\pi_\phi(a'|s')}{\mathbb{E}}[Q_\theta^{\text{new}}(s', a')] + \gamma c \sigma_\theta(s') \ \ . \tag{8}$$

We treat $c$ as a hyperparameter named the **pessimism factor** defined before training and held constant throughout. Appendix F.1 empirically finds that a default value of $c = -0.1$ and details how $c$ may be adjusted. Appendix B.3 shows that the pessimism factor performs the same role as pessimistic ensembles but with more computational efficiency and fine-grained control.

## 3.4 Algorithm

Algorithm 1 presents pseudocode for DoubleGum, an off-policy Deep $Q$-Learning algorithm that learns $\theta$ in both discrete and continuous control. DoubleGum is named after the two Gumbels used in the derivation of its noise model, Equation 4. We outline some implementation details that empirically improved the stability of training.

**Target Networks**: As is standard, $y^{\text{discrete}}$ and $y^{\text{continuous}}$ are computed with target networks, whose parameters we denote by $\psi$. We used target networks with exponential moving weight updating. This

---

**Algorithm 1:** DoubleGum: An Off-Policy Deep $Q$-Learning Algorithm

---

**Input:** MDP $(\mathcal{S}, \mathcal{A}, r, p)$, replay buffer $\mathcal{D} = \emptyset$ (Lin, 1992), initial parameters $\theta_0$ (and $\phi_0$ in continuous control), initial state $s$, learning rates $\eta_\theta$ and $\eta_\phi$, target network EMA rate $\eta_\psi$, batch size $n$.

**Output:** Parameters $\theta$ (and $\phi$ in continuous control)

1 **if** *discrete control* **then**
2     **define** $\pi(s) = \max_a Q_{\theta_i}^{\text{new}}(s, a)$ at iteration $i$
3     **define** $y = y^{\text{discrete}}$
4     **define** $\sigma = \sigma_{\theta_i}(s)$ at iteration $i$
5 **else if** *continuous control* **then**
6     **define** $\pi(s) = \pi_{\phi_i}(s)$ at iteration $i$
7     **define** $y = y^{\text{continuous}}$
8     **define** $\sigma = \sigma_{\theta_i}(s, a)$ at iteration $i$
9 $\psi_0 \leftarrow \theta_0$
10 **for** *iteration $i$* **do**
11     $a \leftarrow \pi(s)$
12     $s' \sim p(s \mid s, a)$
13     $\mathcal{D} \leftarrow \mathcal{D} \cup \{(s, a, r(s, a, s'), s')\}$
14     $s \leftarrow s'$
15     $\theta_{i+1} \leftarrow \theta_i - \eta_\theta \nabla_\theta \underset{(s,a,r,s')_{1:n} \sim \mathcal{D}}{\mathbb{E}} \left[ J_{\theta, y_{\psi_i}, \sigma}(s, a, r, s') \right] \Big|_{\theta_i}$
16     $\psi_{i+1} \leftarrow \eta_\psi \psi_i + (1 - \eta_\psi)\theta_{i+1}$
17     **if** *continuous control* **then**
18         $\phi_{i+1} \leftarrow \phi_i + \eta_\phi \nabla_\phi \underset{(s,a,r,s')_{1:n} \sim \mathcal{D}}{\mathbb{E}} \left[ Q_{\theta_i}^{\text{new}}(s, \pi_\phi(s)) \right] \Big|_{\phi_i}$

---

is commonly used in continuous control (Lillicrap et al., 2015; Fujimoto et al., 2018; Haarnoja et al., 2018b), and has been recently been shown to improve discrete control (D'Oro et al., 2022).

**Q-Functions with Variance Networks**: We learn $\sigma$ with variance networks (Nix and Weigend, 1994; Kendall and Gal, 2017). In discrete control, $\sigma_\theta(s)$ is a variance head added to the $Q$-network, and is approximated as $\sigma_\theta(s, a)$ in continuous control. The variance head is a single linear layer whose input is the penultimate layer of the main $Q$-network and output is a single unit with a SoftPlus activation function (Dugas et al., 2000) that ensures its positivity. Seitzer et al. (2022) improves stability during training by multiplying the entire loss by $\sigma_{\bar{\theta}}$, the numerical value of the variance network, yielding a loss function for $\theta$ which for continuous control is

$$J_{\theta, y_\psi, \sigma_\theta}(s, a, r, s') = \left[ \sigma_{\bar{\theta}}(s, a) \left( \log \sigma_\theta(s, a) + \left( \frac{y_\psi(s, a, r, s') - Q_\theta^{\text{new}}(s, a)}{\sigma_\theta(s, a)} \right)^2 \right) \right] \quad . \quad (9)$$

Note that in the above equation $\sigma_{\bar{\theta}}(s, a)$ would be $\sigma_{\bar{\theta}}(s)$ in discrete control. We do not have any convergence guarantees for this loss function, and Appendix C discusses this issue in more detail.

**Policy**: In discrete control, actions are taken by $\max_a Q_\theta^{\text{new}}(s, a)$ as standard. Exploration was performed by policy churn (Schaul et al., 2022). In continuous control, the policy $\pi_\phi$ is a separate network. We use a DDPG fixed-variance actor because Yarats et al. (2021) showed it trained more stably than a SAC actor with learned variance. Following Laskin et al. (2021), the actor's outputs were injected with zero-mean Normal noise with a standard deviation of 0.2 truncated to 0.3.

**Network Architecture**: All networks had two hidden layers of size 256, ReLU activations (Glorot et al., 2011), orthogonal initialization (Saxe et al., 2013) with a gain of $\sqrt{2}$ for all layers apart from the last layer of the policy and variance head, which had gains of 1. This initialization had been shown to be empirically advantageous in policy-gradient and $Q$-Learning methods in Huang et al. (2022a) and Kostrikov (2021), respectively. Ball et al. (2023) improves stability in continuous control using LayerNorm (Ba et al., 2016). We find that the similar method of GroupNorm (Wu and He, 2018) with 16 groups without a shift and scale (Xu et al., 2019) worked better, but only in continuous control. All parameters were optimized by Adam (Kingma and Ba, 2014) with default hyperparameters.

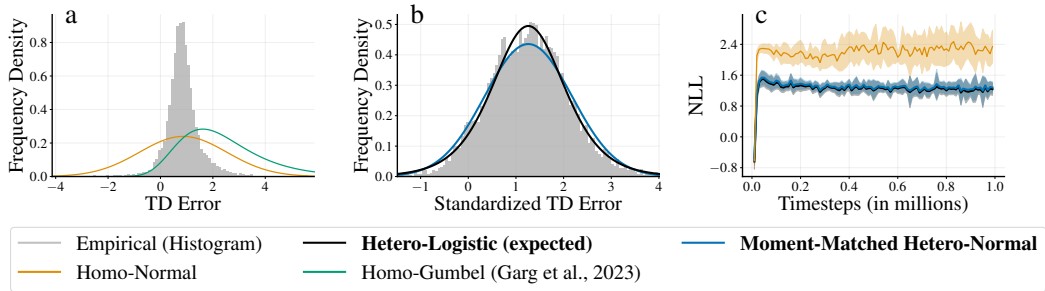

Figure 1: The empirical distribution of noise in discrete control `CartPole-v1`. (a, b): Fitted normal, logistic, and Gumbel distributions against (a) unstandardized and (b) standardized empirical distributions in Deep $Q$-Learning at the end of training. (c): Negative Log-Likelihoods (NLLs) of the noise in Deep $Q$-Learning under different distributions throughout training (lower is better). Appendix D.1 presents further results in more discrete and continuous control tasks.

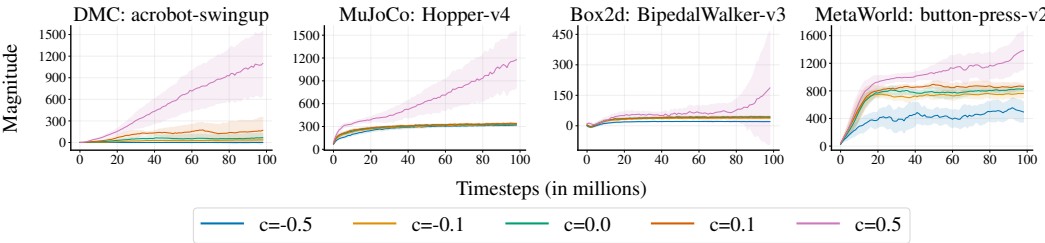

Figure 2: The effect of changing pessimism factor $c$ on the target $Q$-value in continuous control. Appendix D.2 presents further results in more tasks.

## 4 Empirical Evidence for Theoretical Assumptions

### 4.1 Noise Distributions in Deep Q-Learning

Figure 1 shows the evolution of noise during training DoubleGum in the classic control task of `CartPole-v1`. Histograms in Figures 1(a) and 1(b) were generated from $10^4$ samples from the replay buffer after 1 million training timesteps. Figure 1(a) computes TD-errors by $y_\psi(s, a, r, s') - Q_\theta^{\text{new}}(s, a)$, while Figure 1(b) computes standardized TD-errors by dividing the previous equation by $\sigma_\theta(s)$ or $\sigma_\theta(s, a)$ where appropriate. The normal, logistic, and Gumbel distributions were fitted by moment matching, and homo(scedastic) and hetero(scedastic) distributions were respectively fitted to unstandardized and standardized data. Figure 1(c) shows the mean and standard deviation over 12 Negative Log-Likelihoods (NLLs), each computed from a different training run with a different (randomly chosen) initial seed. Every NLL was computed over $10^4$ samples from the replay buffer every 1000 timesteps of training.

Figure 1(a) shows that a homoscedastic normal coarsely fits the empirical distribution, forming a good estimate of the mean but not the spread. Our result contradicts Garg et al. (2023) that fitted a Gumbel to the empirical distribution. We discuss this discrepancy in Appendix E.1. Figure 1(b) shows that a heteroscedastic Logistic captures both the mean and spread of the TD-errors, validating Equations 5 and its derivation. Finally, Figure 1(c) shows that a moment-matched heteroscedastic normal used in DoubleGum is a suitable approximation to the heteroscedastic logistic throughout training, validating Equation 6. Appendix D.1 shows that the trend in Figure 1(c) holds for other discrete control environments as well as continuous control.

### 4.2 Adjusting The Pessimism Factor

Figure 2 plots a 12-sample IQM and standard deviation over $\frac{1}{256} \sum_{i=1}^{256} Q_\psi^{\text{new}}(s_i, a_i)$, the average magnitude of the target $Q$-value used in bootstrapped targets. Figure 2 and Appendix D.2 show that the average magnitude increases as the pessimism factor $c$ increases, validating its effectiveness.

# 5 Related Work

## 5.1 Theoretical Analyses of the Noise in Q-Learning

**Logistic Q-Learning** Bas-Serrano et al. (2021) presents a similar noise model to DoubleGum but with a homoscedastic logistic instead of a heteroscedastic logistic in Equation 5. Their distribution is derived from the linear-programming perspective of value-function learning described in Section 6.9 of Puterman (2014)) and Peters et al. (2010). While DoubleGum learns a $Q$-function off-policy, Logistic $Q$-Learning uses on-policy rollouts to compute the $Q$-values of the current policy. We do not benchmark against Logistic $Q$-Learning because their method is on-policy and only uses linear function approximation.

**Extreme Q-Learning (XQL)** Garg et al. (2023) presents a noise model for Deep $Q$-Learning with one homosecdastic Gumbel noise source, as opposed to the two heteroscedastic Gumbels in Equation 4. Parameters are learned by the LINear-EXponential (LINEX) loss Varian (1975) formed from the log-likelihood of a Gumbel distribution. XQL is presented in more detail in Appendix B.4.

**Gumbel Noise in Deep Q-Learning**: Thrun and Schwartz (1993) argued that the max-operator in bootstrapped targets transformed zero-mean noise from function approximation into statistically biased noise. The authors did not recognize that the resultant noise distribution was Gumbel, and this was realized by Lee and Powell (2012). Unlike DoubleGum, these two works did not make assumptions about the distribution of function approximator noise but instead focused on mitigating the overestimation bias of bootstrapped targets. In economics, McFadden et al. (1973) and Rust (1994) assume the presence of Gumbel noise from noisy reward observations (unlike DoubleGum, which assumes that noise comes from function approximation) and derives the soft value function we present in Appendix A.1 for static and dynamic discrete choice models. XQL brings the Rust-McFadden et al. model to deep reinforcement learning to tackle continuous control.

The **soft value function** was introduced to model stochastic policies (Rummery and Niranjan, 1994; Fox et al., 2015; Haarnoja et al., 2017, 2018b). The most prominent algorithm that uses the soft value function is **Soft Actor-Critic (SAC)** (Haarnoja et al., 2018a,b), which Appendix B.2 shows is a special case of DoubleGum when $Q_\theta^{\text{new}}$ is learned by homoscedastic instead of heteroscedastic normal regression and the spread is a tuned scalar parameter instead of a learned state-dependent standard deviation. Appendix B.2 also shows that **Deep Deterministic Policy Gradients (DDPG)** (Lillicrap et al., 2015) is a simpler special case of SAC that has recently been shown to outperform and train more stably than SAC (Yarats et al., 2021).

## 5.2 Empirically Similar Methods to DoubleGum

**Distributional RL** models the bootstrapped targets and $Q$-function as distributions. In these methods, a $Q$-function is learned by minimizing the divergence between itself and a target distribution. The most similar distributional RL method to DoubleGum is **Mixture-of-Gaussian (MoG) Critics**, introduced in Appendix A of Barth-Maron et al. (2018). DoubleGum is a special case of MoG-Critics with only one Gaussian (such as in Morimura et al. (2012)) and mean samples of the target distribution. Curiously, Shahriari et al. (2022) shows that sample-efficiency of training improves when bootstrapped target sampled are increasingly near the mean but did not try mean sampling. Nevertheless, Shahriari et al. (2022) show that MoG-Critics outperforms a baseline with the C51 distributional head (Bellemare et al., 2017) popular in discrete control, obtaining state-of-the-art results in DeepMind Control. In discrete control with distributional RL, C51 has been superseded by **Quantile Regression (QR)** methods (Dabney et al., 2018b,a; Yang et al., 2019) that predict quantile estimates of a distribution. Ma et al. (2020); Wurman et al. (2022) and Teng et al. (2022) apply QR to continuous control.

**Adjusting pessimism** greatly improves the sample efficiency of RL. Fujimoto et al. (2018) showed empirical evidence of overestimation bias in continuous control and mitigated it with pessimistic estimates computed by **Twin Networks**. However, Twin Networks may sometimes harm sample efficiency because the optimal degree of pessimism varies across environments. To address this, the degree of pessimism is adjusted during training (Lan et al., 2020; Wang et al., 2021; Karimpanal et al., 2021; Moskovitz et al., 2021; Kuznetsov et al., 2020, 2022; Ball et al., 2023), often with the help of an ensemble. In contrast, DoubleGum uses one $Q$-network and one scalar hyperparameter fixed throughout training.

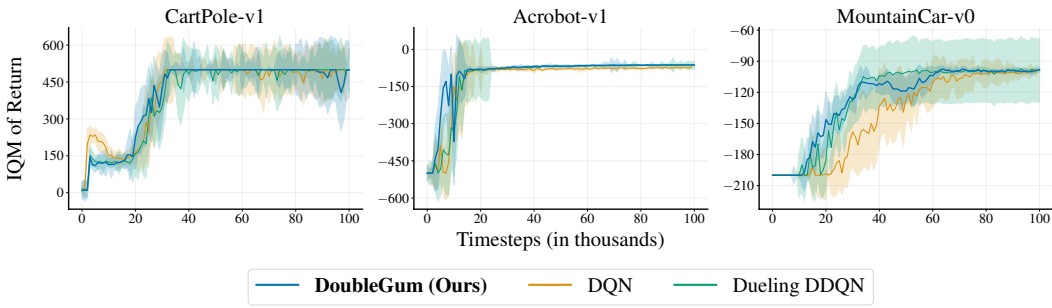

Figure 3: Discrete control, IQM of returns $\pm$ standard deviation over 12 seeds.

Many other RL methods **add or subtract the learned standard deviation to bootstrapped targets**. Risk-Aware RL subtracts a learned standard deviation to learn a low-variance policy (La and Ghavamzadeh, 2013; Tamar and Mannor, 2013; Tamar et al., 2016). Upper-Confidence Bounded (UCB) methods add a learned standard deviation to explore high-variance regions (Lee et al., 2021; Teng et al., 2022). These methods use a combination of ensembles and variance networks, but it is also possible to derive a Bellman equation for the variance following Dearden et al. (1998). The current state-of-the-art method in RL with variance estimation is Mai et al. (2022), which uses both variance networks and ensembles. The use of ensembles is motivated by the need to capture model uncertainty – differences between $Q$-functions with different parameters. We believe that ensembles are unnecessary because model uncertainty will be expressed in bootstrapped targets from two different timesteps as parameters change through learning and that all variation in bootstrapped targets will henceforth be captured by variance networks.

## 6 Results

### 6.1 Discrete Control

We benchmarked DoubleGum on classic discrete control tasks against two baselines from the DQN family of algorithms (Figure 3). All algorithms were implemented following Section 3.4. Appendix E.3 discusses the baseline algorithms in more detail.

Performance was evaluated by the InterQuartile Mean (IQM) over 12 runs, each one with a different randomly initialized seed. Agarwal et al. (2021) showed that the IQM was a robust performance metric in RL. 12 was chosen because it was the smallest multiple of four (so a IQM could be computed) greater than the 10 seeds recommended by Henderson et al. (2018). In each run, the policy was evaluated by taking the mean of 10 rollouts every 1000 timesteps, following Fujimoto et al. (2018).

Figure 3 shows that DoubleGum sometimes obtains better sample efficiency than baselines, but not significantly more to necessitate further discrete control experiments. In the remainder of this work, we focus on continuous control, where we found DoubleGum to be more effective.

### 6.2 Continuous Control

We present two modes of evaluating DoubleGum because our algorithm has a pessimism factor hyperparameter we choose to tune per suite. We, therefore, benchmark all continuous control algorithms with default pessimism (Figure 4) and without pessimism-tuning per-suite (Figure 5). Table 5 presents default and per-suite pessimism values.

We compare against seven baselines. The first five are popular algorithms in the continuous control literature: DDPG (Lillicrap et al., 2015), TD3 (Fujimoto et al., 2018), SAC (Haarnoja et al., 2018b), MoG-Critics (Shahriari et al., 2022) and XQL (Garg et al., 2023). Here, DDPG, TD3, and SAC represent MaxEnt RL, MoG-Critics represent Distributional RL, and XQL is the algorithm with the most similar noise model to ours. We introduce two further baselines: QR-DDPG, a stronger Distributional RL baseline that combines quantile regression (Dabney et al., 2018b) with DDPG, and FinerTD3, TD3 with an ensemble of five networks as opposed to a network of two ensembles

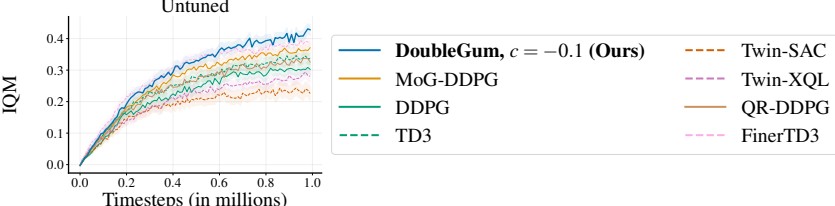

Figure 4: Continuous control with default parameters, IQM normalized score over 33 tasks in 4 suites with 95% stratified bootstrap CIs. Methods that default to use Twin Networks are dashed. Appendix F.4 presents per-suite and per-task aggregate results.

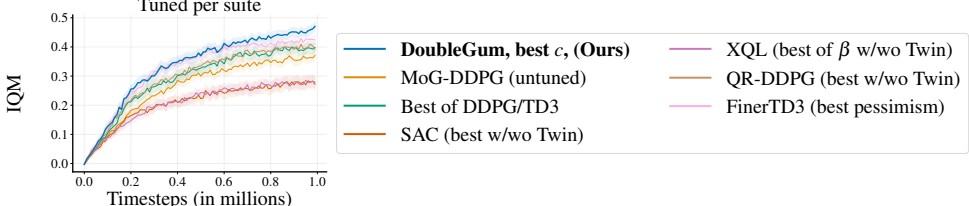

Figure 5: Continuous control with the best pessimism hyperparameters tuned per suite, IQM normalized score over 33 tasks in 4 suites with 95% stratified bootstrap CIs. Appendix F.5 presents per-suite and per-task aggregate results.

in the original TD3 that enables finer control over the degree of pessimism with an ensemble of five networks. All algorithms were implemented following design decisions outlined in Section 3.4. Appendix E.4 discusses the algorithms in more detail.

The pessimism of DoubleGum was tuned by changing its pessimism factor. The pessimism of baseline algorithms was tuned by manually choosing whether to use Twin Networks (Fujimoto et al., 2018) or not. Note that we refer to DDPG with Twin Networks as TD3. The pessimism of FinerTD3 was tuned by selecting which quantile estimate to use from an ensemble of five networks. The pessimism of MoG-Critics could not be tuned because its critic does not support paired sampling between ensemble members. Appendices F.1 and F.2 respectively detail how the pessimisms of DoubleGum and baseline algorithms were adjusted.

We benchmarked DoubleGum on 33 tasks over 4 continuous control suites comprising respectively of 11 DeepMind Control (DMC) tasks (Tassa et al., 2018; Tunyasuvunakool et al., 2020), 5 MuJoCo tasks (Todorov et al., 2012; Brockman et al., 2016), 15 MetaWorld tasks (Yu et al., 2020) and 2 Box2D tasks (Brockman et al., 2016). We follow Agarwal et al. (2021) and evaluate performance with the normalized IQM with 95% stratified bootstrap confidence intervals aggregated over all 33 tasks. 12 runs from each task was collected by a similar method to that described in Section 6.1. Further details of the tasks and their aggregate metric are detailed in Appendix E.2.

Figure 4 shows that DoubleGum outperformed all baselines in aggregate over 33 tasks when all algorithms used their default pessimism settings. Figure 5 shows that DoubleGum outperformed all baselines in aggregate over 33 tasks when the pessimism of all algorithms is adjusted per suite. Comparing the figures shows that adjusting the pessimism of DoubleGum per suite also attained a higher aggregate score than DoubleGum with default pessimism.

## 7 Discussion

This paper studied the noise distribution in Deep $Q$-Learning from first principles. We derived a noise model for Deep $Q$-Learning that used two heteroscedastic Gumbel distributions. Converting our noise model into an algorithm yielded DoubleGum, an off-policy Deep $Q$-Learning algorithm applied to both discrete and continuous control.

In discrete control, our algorithm attained competitive performance to the baselines. Despite having an numerically exact loss function in discrete control, DoubleGum was very sensitive to hyperparameters.

Practically, using Dueling DQN (Wang et al., 2016) to learn a $Q$-function was crucial to getting DoubleGum to work. Appendix D.1 shows that our noise model fits the underlying noise distribution of Deep $Q$-Learning and we therefore suspect that instability in discrete control might be due to the training dynamics of deep learning and not our theory.

In continuous control, we introduced a pessimism factor hyperparameter to approximate our otherwise intractable noise model. We provided a default value for the pessimism factor that outperformed popular $Q$-Learning baselines in aggregate over 33 tasks. Tuning this hyperparameter yielded even greater empirical gains. Our method of tuning pessimism was more computationally efficient and finer-grained than popular methods that tuned a quantile estimate from an ensemble.

In continuous control, DoubleGum outperformed all baselines in aggregate. We hypothesize that DoubleGum outperformed MaxEnt RL baselines because DoubleGum is a more expressive generalization of SAC, which is itself more expressive than DDPG as shown in Appendix B.2. Our theory showed that TD-errors follow a heteroscedastic Logistic, and we believe that modeling this distribution should be sufficient for distributional RL. We hypothesize that more complex distributions considered by the Distributional RL methods QR and MoG overfit to the replay buffer and might not generalize well to online rollouts. FinerTD3 performs marginally poorer than DoubleGum, even when pessimism was adjusted per-suite. We believe this is because FinerTD3 adjusts pessimism finer than other baselines, but still not as fine as the continuous scalar in DoubleGum. Finally, we hypothesize that DoubleGum outperformed XQL because our noise model better fits the underlying noise distribution in Deep $Q$-Learning, as shown in Appendix D.1.

This paper shows that better empirical performance in Deep RL may be attained through a better understanding of theory. To summarize, we hope that our work encourages the community to increase focus on reducing the gap between theory and practice to create reinforcement learning algorithms that train stably across a wide variety of environments.

## 7.1 Limitations

Theoretically, our work lacks a convergence guarantee for DoubleGum. This is exceptionally challenging because to the best of our knowledge there are currently no convergence guarantees for heteroscedastic regression. Appendix C discusses convergence in more detail.

Experimentally, there are many areas left open for future work. For speed in proof of concept experiments, we only swept over five values for the pessimism factor hyperparameter and only tuned per-suite. We anticipate that a more thorough search of the pessimism factor combined with a per-task selection will improve our results even further. An obvious follow-up would be an algorithm that automatically adjusted the pessimism factor during training. Additionally, we only focused on environments with state observations to not deal with representation learning from visual inputs. Another obvious next step would be to train DoubleGum on visual inputs or POMDPs.

## Acknowledgements

We would like to thank Clement Gehring, Victor Hui, Sobhan Mohammadpour, Tianwei Ni, Anushree Rankawat, Fabrice Normandin, Ryan D'Orazio, Jesse Farebrother, Valentin Thomas, Matthieu Geist, Olivier Pietquin, Emma Frejinger, and the five anonymous reviewers.

All algorithms and code were implemented in JAX (Bradbury et al., 2018) and Haiku (Hennigan et al., 2020) and cpprb (Yamada, 2019). Helper functions for running RL experiments were taken from Kostrikov (2021); Wu (2021); Hoffman et al. (2020) and Agarwal et al. (2021). Compute requirements are listed in Appendix E.5.

This research was funded by CIFAR and Hitachi and enabled in part by compute resources, software and technical help provided by Mila (`mila.quebec`), Calcul Québec (`calculquebec.ca`) and the Digital Research Alliance of Canada (`alliance.can.ca`).

Hui carried out the majority of his work in Section J of Mila and would like to express his wish for the silent (but vibrant) working conditions there to continue for many years to come.

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

# Appendix

## Table of Contents

# A Proofs

## A.1 The SoftMax Location of Gumbel Distribution

We refer to the $\log$-sum-$\exp$ operator as the SoftMax operator. This is not the same-named operator in Bridle (1989, 1990), which we suggest should be (re-)named SoftArgMax.

**Theorem 1.**

$$\max_i[\alpha_i + g_i] = \beta \log \sum_i \exp\left(\frac{\alpha_i}{\beta}\right) + g, \quad where \ \ g, g_i \sim \mathcal{G}(0, \beta)$$

*where $\mathcal{G}$ is a Gumbel distribution. A Gumbel random variable $g \sim \mathcal{G}(\alpha, \beta)$ specified by location $\alpha \in \mathbb{R}$ and spread $\beta > 0$ has PDF $p(g) = \frac{1}{\beta} \exp(-z - \exp(-z))$ with $z = (g - \alpha)/\beta$ and CDF $P(g) = \frac{1}{\beta} \exp(-\exp(-z))$ (Gumbel, 1935).*

*Proof.* First note

$$\alpha_i + g_i, \ \ g_i \sim \mathcal{G}(0, \beta) \implies \alpha_i + g_i \sim \mathcal{G}_i(\alpha_i, \beta)$$

Then, denoting $y = \max_i[\alpha_i + g_i]$

$$\begin{aligned}
P(X \leqslant y) &= \prod_i P_i(X \leqslant y) \\
&= \prod_i \mathcal{G}_i(X \leqslant y) \\
&= \prod_i \exp\left(-\exp\left(-\frac{x - \alpha_i}{\beta}\right)\right) \\
&= \exp\left(-\sum_i \exp\left(-\frac{x - \alpha_i}{\beta}\right)\right) \\
&= \exp\left(-\exp\left(-\frac{x}{\beta}\right) \sum_i \exp\left(\frac{\alpha_i}{\beta}\right)\right) \\
&= \exp\left(-\exp\left(-\frac{x}{\beta} + \log \sum_i \exp\left(\frac{\alpha_i}{\beta}\right)\right)\right) \\
&= \exp\left(-\exp\left(-\frac{1}{\beta}\left(x - \beta \log \sum_i \exp\left(\frac{\alpha_i}{\beta}\right)\right)\right)\right) \\
&= \mathcal{G}\left(\beta \log \sum_i \exp\left(\frac{\alpha_i}{\beta}\right), \beta\right) \\
&= \beta \log \sum_i \exp\left(\frac{\alpha_i}{\beta}\right) + g, \quad where \ \ g \sim \mathcal{G}(0, \beta)
\end{aligned}$$

$\square$

When applied to discrete $Q$-Learning, we produce

$$\max_{\mathcal{A}} [Q_\theta(s, a) + g_{\theta, a}(s)] = \beta_\theta(s) \log \int_{\mathcal{A}_s} \exp\left(\frac{Q_\theta(s, a)}{\beta_\theta(s)}\right) \mathrm{d}a + g_\theta(s) \tag{10}$$

where $g_{\theta, a}(\cdot), g_\theta(\cdot) \sim \mathcal{G}(0, \beta_\theta(\cdot))$, for all $a \in \mathcal{A}$.

We assume that the same result holds for the continuous case if $|\mathcal{A}| < \infty$, an assumption first used in Lemma 1, Appendix B.1, Page 11 of Haarnoja et al. (2018a) to ensure boundedness. Here, we require the output of the $\max$-operator to be bounded, which cannot be the case when the number of its arguments is $\infty$.

## A.2 Shifting the Value Function

**Theorem 2.**

$$\beta \log \sum_i \exp\left(\frac{\alpha_i + g}{\beta}\right) = \beta \log \sum_i \exp\left(\frac{\alpha_i}{\beta}\right) + g$$

*Proof.*

$$\beta \log \sum_i \exp\left(\frac{\alpha_i + g}{\beta}\right) = \beta \log \sum_i \exp\left(\frac{\alpha_i}{\beta} + \frac{g}{\beta}\right) \tag{11}$$

$$= \beta \log \exp\left(\frac{g}{\beta}\right) \sum_i \exp\left(\frac{\alpha_i}{\beta}\right) \tag{12}$$

$$= \beta \log \exp\left(\frac{g}{\beta}\right) + \beta \log \sum_i \exp\left(\frac{\alpha_i}{\beta}\right) \tag{13}$$

$$= \beta \log \sum_i \exp\left(\frac{\alpha_i}{\beta}\right) + g \tag{14}$$

$\square$

## A.3 The Difference between Two Gumbel Random Variables is a Logistic

**Theorem 3.**

$$x_1, x_2 \sim \mathcal{G}(0, \beta) \implies z = x_1 - x_2, \quad z \sim \mathcal{L}(0, \beta)$$

*where $\mathcal{L}$ is a logistic distribution. A logistic random variable $l \sim L(\alpha, \beta)$ with location $\alpha$ and spread $\beta$ has PDF $\frac{\exp\left(-(l-\alpha)/\beta\right)}{\beta(1+\exp\left(-(l-\alpha)/\beta\right))^2}$ and CDF $\frac{1}{1+\exp\left(-(l-\alpha)/\beta\right)}$.*

*Proof.* First, construct the convolution based on the joint PDF

$$
\begin{aligned}
\mathcal{L}(z) &= P(Z \leqslant z) \\
&= P(X_1 - X_2 \leqslant z) \\
&= P(X_1 \leqslant z + x_2) \\
&= \int_{-\infty}^{\infty} \int_{-\infty}^{z+x_2} g(x_1)\, g(x_2)\, \mathrm{d}x_1\, \mathrm{d}x_2 \\
&= \int_{-\infty}^{\infty} G(z + x_2)\, g(x_2)\, \mathrm{d}x_2 \ .
\end{aligned}
$$

Rewriting $x_2$ as $x$ yields

$$
\begin{aligned}
\mathcal{L}(z) &= \int_{-\infty}^{\infty} G(z + x)\, g(x)\, \mathrm{d}x \\
&= \int_{-\infty}^{\infty} \exp\left(\exp-\frac{z+x}{\beta}\right) \exp\left(-\frac{x}{\beta} - \exp-\frac{x}{\beta}\right) \mathrm{d}x \\
&= \int_{-\infty}^{\infty} \exp\left(-e^{-\frac{x}{\beta}}\left(1 + e^{-\frac{z}{\beta}}\right)\right) e^{-\frac{x}{\beta}}\, \mathrm{d}x \\
&= \int_0^{\infty} \frac{1}{\beta} \exp\left(-u\left(1 + e^{-\frac{z}{\beta}}\right)\right) \mathrm{d}u, \quad \text{where } u = e^{-\frac{x}{\beta}},\ \mathrm{d}u = -\frac{1}{\beta}e^{-\frac{x}{\beta}}\mathrm{d}x \\
&= \frac{1}{\beta}\frac{1}{1 + e^{-\frac{z}{\beta}}} e^{\left(-u\left(1 + e^{-\frac{z}{\beta}}\right)\right)}\bigg|_0^{\infty} \\
&= \frac{1}{1 + e^{-\frac{z}{\beta}}} \ .
\end{aligned}
$$

which is the CDF of $\mathcal{L}(0, \beta)$. $\square$

## A.4 Soft Q-Learning Identity

**Theorem 4.** *For an arbitrary $p(x)$*

$$\beta \log \int \exp\left(\frac{E(x)}{\beta}\right) dx = \mathop{\mathbb{E}}_{x \sim p(\cdot)} [E(x)] + \beta\, \mathbb{C}[p \,||\, p^\star], \;\; where \;\; p^\star(x) = \frac{\exp\frac{E(x)}{\beta}}{\int \exp\frac{E(x)}{\beta} dx}$$

*Proof.*

$$\mathop{\mathbb{E}}_{p(x)} [E(x)] + \beta\, \mathbb{C}[p \,||\, p^\star] = \mathop{\mathbb{E}}_{p(x)} [E(x)] - \beta \int p(x) \log \frac{\exp\frac{E(x)}{\beta}}{\int \exp\frac{E(x')}{\beta} dx'} dx$$

$$= \beta \int p(x) \log \int \exp\frac{E(x')}{\beta} dx' \, dx$$

$$= \beta \log \int \exp\frac{E(x)}{\beta} dx$$

$\square$

When applied to $Q$-Learning, the following identity produces

$$\beta(s) \log \int \exp\left(\frac{Q_\theta(s,a)}{\beta(s)}\right) da = \mathop{\mathbb{E}}_{\pi_\phi(a'|s')} [Q_\theta(s,a)] + \beta(s)\, \mathbb{C}[\pi_\phi \,||\, p_\theta]$$

$$where \;\; p_\theta(a \mid s) = \frac{\exp\frac{Q_\theta(s,a)}{\beta(s)}}{\int \exp\frac{Q_\theta(s,a')}{\beta(s)} da'}$$

# B  Further Theory and Derivations

## B.1  Actor Loss

The actor losses used in DoubleGum, SAC, and DDPG are all derived from the same principle. For a given $s$, the actor loss function should minimizes the following (reverse) KL-Divergence, previously presented in Equation 7.

$$\min_\phi \beta\, \mathbb{D}_{\mathrm{KL}}[\pi_\phi \,||\, p_\theta], \;\; where \;\; p_\theta(a \mid s) = \frac{\exp Q_\theta^{\mathrm{new}}(s,a)/\beta}{\int \exp Q_\theta^{\mathrm{new}}(s,a')/\beta\, da'} \;\;.$$

This simplifies as

$$\min_\phi \beta\, \mathrm{D}_{\mathrm{KL}}[\pi_\phi \,||\, p_\theta] = \min_\phi \beta \int \pi_\phi(a \mid s) \log \frac{\pi_\phi(a \mid s)}{p_\theta(a \mid s)} da$$

$$= \max_\phi \left[ \beta\, \mathbb{H}[\pi_\phi] + \beta \int \pi_\phi(a \mid s) \log \frac{\exp Q_\theta^{\mathrm{new}}(s,a)/\beta}{\int \exp Q_\theta^{\mathrm{new}}(s,a')/\beta\, da'} da \right]$$

$$= \max_\phi \left[ \beta\, \mathbb{H}[\pi_\phi] + \beta \int \pi_\phi(a \mid s) \frac{Q_\theta^{\mathrm{new}}(s,a)}{\beta} da \right]$$

which is then estimated by Monte-Carlo samples from $\pi_\phi$ as

$$\max_\phi \mathop{\mathbb{E}}_{\pi_\phi(a|s)} [Q_\theta^{\mathrm{new}}(s,a) - \beta \log \pi_\phi(a \mid s)] \;\;. \tag{15}$$

SAC (Haarnoja et al., 2018a,b) has a policy with learned variance and state-independent $\beta$. DDPG (Lillicrap et al., 2015) has a fixed-variance policy which removes the second term in Equation 15 as it is constant with respect to the maximization. DoubleGum has a state-dependent $\beta(s)$, but uses the same actor loss as DDPG because DoubleGum uses a DDPG fixed-variance policy.

## B.2 Maximum-Entropy Reinforcement Learning

SACv1 (Haarnoja et al., 2018b) is a special case of DoubleGum and DDPG (Lillicrap et al., 2015) is a special case of SAC. All three continuous control algorithms have an actor and critic loss derived from the same principle. Section B.1 shows this for the actor losses of DoubleGum, SAC, and DDPG. We now relate the critic losses to each other, starting from the most general case, DoubleGum. In continuous control, DoubleGum uses the following noise model, formed from substituting Equation 7 into Equation 5:

$$Q_\theta^{\text{new}}(s, a) + l_{\theta,a}(s) = \mathop{\mathbb{E}}_{p(s'|s,a)} \left[ r + \gamma \mathop{\mathbb{E}}_{\pi_\phi(a'|s')} [Q_\theta^{\text{new}}(s', a')] + \gamma \beta_\theta(s) \, \mathbb{C}[\pi_\phi \| p_\theta] \right] \ . \tag{16}$$

Here, $l_{\theta,a}(\cdot) \sim \mathcal{L}(0, \beta_\theta(\cdot))$ is a logistic distribution and $p_\theta(a \mid s) \propto \exp \frac{Q_\theta^{\text{new}}(s,a)}{\beta_\theta(s)}$. The DoubleGum critic loss is derived from this noise model by approximating the RHS with Equation 8 and learning $\theta$ with moment matching in Section 3.2.

The SAC noise model is derived from Equation 16 in three ways. First, SAC approximates $l_{\theta,a}(s) \sim \mathcal{L}(0, \beta_\theta(\cdot))$ as $n_{\theta,a} \sim \mathcal{N}(0, \sigma)$, motivated by the fact that both distributions have the same mean/mode. Secondly, SAC approximates the DoubleGum state-dependent logistic spread $\beta_\theta(s)$ as temperature parameter $\beta$ learned not as a part of the critic but by itself with Lagrangian dual gradient descent. Thirdly, SAC breaks down $\mathbb{C}[\pi_\phi \| p_\theta] = \mathbb{H}[\pi_\phi] + \mathbb{D}_{\text{KL}}[\pi_\phi \| p_\theta]$ before assuming that the KL-Divergence is negligible, given that it is minimized by the actor loss. These three approximations yield the SAC noise model as

$$Q_\theta^{\text{new}}(s, a) + n_{\theta,a} = \mathop{\mathbb{E}}_{p(s'|s,a)} \left[ r + \gamma \mathop{\mathbb{E}}_{\pi_\phi(a'|s')} [Q_\theta^{\text{new}}(s', a') + \beta \log \pi_\phi(a' \mid s')] \right] \ . \tag{17}$$

MLE of $\theta$ wrt the above noise model yields the MSBE critic loss.

DDPG is a special case of SAC that assumes $\beta \to 0$, removing the last term in Equation 17. $\lim_{\beta \to 0} p_\theta(a \mid s)$ becomes deterministic, so $\pi_\phi$ may be modelled by a deterministic policy.

## B.3 Interpreting the Cross-Entropy as a Pessimism Factor

In continuous control, Fujimoto et al. (2018) introduced Twin Networks, a method that improved sample-efficiency with pessimistic bootstrapped targets computed by returning a sample-wise minimum from an ensemble of two $Q$-functions. Follow-up work selects a quantile estimate from an ensemble (Kuznetsov et al., 2020; Chen et al., 2021; Ball et al., 2023), which we demonstrate is equivalent to estimating $V_{\theta,\beta}^{\text{soft, new}}$.

Suppose there is an ensemble of $n$ networks where the i[th] network follows $Q_{\theta_i}(s, a) = Q_\theta(s, a) + z_i(s, a)$. Here, $Q_\theta$ is an 'ideal' function approximator never instantiated nor computed and $z$ is an arbitrary noise source. When $n$ is sufficiently large,

$$\min_i \mathop{\mathbb{E}}_{\pi_\phi(a|s)} [Q_{\theta_i}(s, a)] = \min_i \mathop{\mathbb{E}}_{\pi_\phi(a|s)} [Q_\theta(s, a) + z_i(s, a)] = \mathop{\mathbb{E}}_{\pi_\phi(a|s)} [Q_\theta(s, a)] + \min_i z_i(s)$$

$$= \mathop{\mathbb{E}}_{\pi_\phi(a|s)} [Q_\theta(s, a)] - g(s), \quad \text{where } g(s) \sim \mathcal{G}(\alpha(s), \beta(s)) \ .$$

A Gumbel random variable $g \sim \mathcal{G}(\alpha, \beta)$ has $\mathbb{E}[g] = \alpha + \gamma_e \beta$, where $\gamma_e$ is the Euler-Mascheroni constant, so for a deterministic environment the bootstrapped targets become

$$r + \gamma \mathop{\mathbb{E}}_{\pi_\phi(a'|s')} [Q_\theta(s', a')] - \gamma \alpha(s') - \gamma \gamma_e \beta(s') \ ,$$

recovering Equation 8, the DoubleGum continuous control targets, up to an additive term $\gamma \alpha(s')$, while $-\gamma \gamma_e \beta(s')$ recovers the spread $\gamma c \sigma_\theta(s')$ up to a negative scaling factor, indicating that the default $c$ should be negative. Moskovitz et al. (2021) and Ball et al. (2023) showed that the appropriate ensemble size and selected quantile changes the overestimation bias, so appropriate values would ensure $\alpha(s') = 0$.

### B.4 Comparison between DoubleGum and XQL

We present an explanation of Extreme $Q$-Learning (XQL) as presented in Appendix C.1 of Garg et al. (2023). XQL can be derived from Soft Bellman Equation backups given by

$$Q(s,a) \leftarrow \mathop{\mathbb{E}}_{p(s'|s,a)}[r(s,a,s') + \gamma V^{\text{soft}}(s)], \quad \text{where} \quad V^{\text{soft}}(s) = \beta \log \sum_{a'} \exp\left(\frac{Q(s',a')}{\beta}\right)$$

and $\beta$ is a fixed hyperparameter. Computing the log-sum-exp of $V^{\text{soft}}$ is intractable in continuous control, as the sum over $a'$ becomes an integral in continuous control tasks.

Garg et al. (2023) present a method of estimating its value using Gumbel regression. Given a (potentially infinite) set of scalars $x \in X$, Gumbel regression provides a method to estimate the numerical value of $\log\text{-sum-exp}_\beta(x) = \beta \log \sum_X \exp x/\beta$. Gumbel regression assumes $x \sim \mathcal{G}(\alpha,\beta)$, where $\mathcal{G}$ is a homoscedastic Gumbel distribution, and $\beta$ is a fixed (hyper)parameter. $\alpha$ estimated by MLE tends towards $\log\text{-sum-exp}_\beta(x)$. MLE is performed by numerically maximizing the $\log$-likelihood of a Gumbel distribution, which recovers the LINear-EXponential (LINEX) loss function introduced by Varian (1975).

Garg et al. (2023) incorporate Gumbel regression into deep $Q$-Learning in two ways, which they name X-SAC and X-TD3. X-SAC combines Gumbel regression to estimate the soft value function used in SACv0 (Haarnoja et al., 2018a). The soft value function $V_\rho^{\text{soft}}(s)$ is a neural network whose parameters $\rho$ are learned by Gumbel regression from $Q_\psi(s,a) \sim \mathcal{G}(V_\rho(s),\beta)$, where $\psi$ are target network parameters. A neural network $Q_\theta$ with parameters $\theta$ may then be learned by the MSE between itself and $\mathbb{E}_{p(s'|s,a)}[r(s,a,s') + \gamma V_\rho^{\text{soft}}(s)]$. X-SAC is vastly different from DoubleGum, because our algorithm does not estimate the soft value function with a separate neural network.

Gumbel regression is directly used to learn the $Q$-values in X-TD3. First, the bootstrapped targets are thusly rewritten

$$y^{\text{soft}}(s,a) = \mathop{\mathbb{E}}_{p(s'|s,a)}\left[r + \gamma\beta \log \sum_{a'} \exp\left(\frac{Q_\phi(s',a')}{\beta}\right)\right]$$

$$= \mathop{\mathbb{E}}_{p(s'|s,a)}\left[\gamma\beta \log \sum_{a'} \exp\left(\frac{r + \gamma Q_\psi(s',a') - Q_\theta(s,a)}{\gamma\beta}\right)\right]$$

In environments with deterministic environments, which comprise all environments considered by Garg et al. (2023) and our paper, Lemma C.1 of Garg et al. (2023) provides a method of learning the soft value function with Gumbel regression on $y^{\text{soft}}(s,a) \sim \mathcal{G}(Q_\theta(s,a),\gamma\beta)$. The Gumbel regression objective used in X-TD3 to learn $\theta$ is vastly different from the moment matching with the logistic distribution DoubleGum uses to learn $\theta$.

To motivate their use of Gumbel regression, Garg et al. (2023) derived a noise model which they use to present empirical evidence of homoscedastic Gumbel noise. In contrast, we presented empirical evidence of heteroscedastic logistic noise formed from a noise model with two heteroscedastic Gumbel distributions.

## C  A Discussion on The Convergence of DoubleGum

To the best of our knowledge, there are two types of convergence analysis in $Q$-Learning: 1) operator-theoretic analysis over tabular $Q$-functions, and 2) training dynamics of neural network parameters. We believe the second is more appropriate for DoubleGum, because our theory addresses issues in using neural networks (and not tables) for $Q$-learning. Nevertheless, for completeness, we discuss convergence guarantees for the tabular setting and the function approximation setting. While we can guarantee convergence for the former setting, we have no guarantees for the second.

### C.1 Tabular Q-Functions

Appendices B.1 and B.2 present DoubleGum as a MaxEnt RL algorithm. When $Q$-functions are tabular, Appendix A of Haarnoja et al. (2018a) shows that MaxEnt RL algorithms may be derived

**Algorithm 2:** DoubleGum Soft Policy Iteration

---

**Input:** Finite MDP $(\mathcal{S}, \mathcal{A}, r, p)$, initial tables $Q$, $\beta$, $y^{\text{soft}}$, initial policy $\pi$
**Output:** Optimal Tabular $Q$-function $Q^{\star}$

1  **for** *training iteration $i$* **do**
2      **for** *all $s$* **do**
3          **for** *all $a$* **do**
4              $y^{\text{soft}}(s,a) \leftarrow \mathop{\mathbb{E}}\limits_{p(s'|s,a)}\left[ r(s,a,s') + \gamma\beta_i(s')\log\sum_{a'}\exp\left(\frac{Q_i(s',a')}{\beta_i(s')}\right)\right]$
5              $Q_{i+1}(s,a) \leftarrow y^{\text{soft}}(s,a)$
6              $\beta_{i+1}(s) \leftarrow \frac{\sqrt{3}}{\pi}\sqrt{\mathop{\mathbb{V}}\limits_{a\sim\pi(a|s)}[y^{\text{soft}}(s,a)]}$
7              **define** $\pi(a\mid s) \leftarrow \frac{\exp(Q_{i+1}(s,a)/\beta_{i+1}(s))}{\sum_{a'}\exp(Q_{i+1}(s,a')/\beta_{i+1}(s))}$

---

from soft policy iteration. We therefore present a convergence proof for DoubleGum with tabular $Q$-functions based on soft policy iteration.

DoubleGum treats the return as coming from a logistic distribution and learns its location and spread. In the tabular setting, two tables would need to be learned, $Q(s,a)$ and $\beta(s)$. An algorithm to learn these tables in a finite MDP with soft policy iteration is presented in Algorithm 2. Policy evaluation is done by Lines 4-6 while Line 7 performs policy improvement.

Proof of convergence of Algorithm 2 is similar to the SAC proof of convergence in Appendix B of Haarnoja et al. (2018a). This should not be surprising, given that Appendix B.2 shows SAC as a special case of DoubleGum. We first show that policy evaluation converges and that a new policy found by policy improvement does not reduce the magnitude of the value function.

**Lemma 5** (Soft Policy Evaluation). *Consider the Soft Policy Evaluation operator given by*

$$Q_{i+1}(s,a) \leftarrow \mathop{\mathbb{E}}\limits_{p(s'|s,a)}\left[ r(s,a,s') + \gamma\beta_i(s')\log\sum_{a'}\exp\left(\frac{Q_i(s',a')}{\beta_i(s')}\right)\right] \quad \text{over all } (s,a) \text{ pairs.}$$

$\lim_{i\to\infty} Q_i$ *converges to the soft Q-value.*

*Proof.* Following Appendix A.4

$$\beta(s)\log\sum_a\exp\left(\frac{Q(s,a)}{\beta(s)}\right) = \mathop{\mathbb{E}}\limits_{\pi(a|s)}[Q_\theta(s,a)] + \beta(s)\,\mathbb{C}[\pi\,||\,p]$$

$$\text{where } p(a\mid s) = \frac{\exp(Q(s,a)/\beta(s))}{\sum_{a'}\exp(Q(s,a')/\beta(s))}$$

the bootstrapped targets may be thusly rewritten

$$\mathop{\mathbb{E}}\limits_{p(s'|s,a)}\left[ r(s,a,s') + \gamma\beta(s')\log\sum_{a'}\exp\left(\frac{Q(s',a')}{\beta(s')}\right)\right]$$

$$= \mathop{\mathbb{E}}\limits_{p(s'|s,a)}\left[ r(s,a,s') + \gamma\mathop{\mathbb{E}}\limits_{\pi(a'|s')}[Q(s',a')] + \beta(s')\,\mathbb{C}[\pi\,||\,p]\right]$$

$$= \mathop{\mathbb{E}}\limits_{p(s'|s,a)}\left[ r'(s,a,s') + \gamma\mathop{\mathbb{E}}\limits_{\pi(a'|s')}[Q(s',a')]\right]$$

where $r'(s,a,s') = r(s,a,s') + \beta(s')\,\mathbb{C}[\pi\,||\,p]$.

Following Lemma 1 in Haarnoja et al. (2018a), Sutton and Barto (1998) gives convergence of $Q_{i+1}(s,a) \leftarrow \mathbb{E}_{p(s'|s,a)}\left[r'(s,a,s') + \gamma\mathbb{E}_{\pi(a'|s')}[Q_i(s',a')]\right]$         $\square$

The proof of Soft Policy Improvement should be identical to SAC, given that Appendix B.1 shows that DoubleGum and SAC use identical actor losses. As such, Lemma 5 can be used in place of Lemma 1 in Theorem 1 of Haarnoja et al. (2018a), thus showing convergence of DoubleGum in the tabular setting.

## C.2 Deep Q-Functions

Parameters of the deep $Q$-function used by DoubleGum in Algorithm 1 are learned by a loss function equivalent to that of heteroscedastic normal regression. Convergence of DoubleGum in the function approximation setting would therefore rely on convergence of heteroscedastic normal regression.

Zhang et al. (2023) introduces PAC-bounds for heteroscedastic normal regression, but on the condition that the mean-estimate is close to the ground truth mean, as mentioned in Paragraph 1 of Section 4. This is empirically achieved by Seitzer et al. (2022), who analyze heteroscedastic normal regression and find that the mean-estimate frequently converges to an underfitting solution. This is because the Negative Log-Likelihood (NLL) of a normal distribution is minimized when the variance becomes large – in Equation 6, this term is denoted with $\sigma_\theta^2$. As such, changes in $Q_\theta^{\text{new}}$ will not change the loss function much. To rectify this, Seitzer et al. (2022) multiplies the NLL of the normal with the numerical value of the standard deviation, reducing the dominance of $\sigma_\theta$ on the loss function.

# D Further Empirical Evidence for Theoretical Assumptions

## D.1 Noise Distributions in Deep Q-Learning

Figure 6 presents graphs corresponding to Figure 1c for all environments considered in this paper. Continuous control results were generated from DoubleGum with default pessimism ($c = -0.1$).

## D.2 Adjusting The Pessimism Factor

Figure 7 presents graphs corresponding to Figure 2 for all continuous control environments considered in this paper.

# E Further Experimental Details

## E.1 Noise Distribution Discrepancy with Extreme Q-Learning

In Appendix D.2 of Page 19, Garg et al. (2023) fitted a Gumbel distribution to the TD errors on three continuous control environments. The Gumbel distribution was a good fit in two of the three environments they benchmarked on. We could not reproduce this result and attribute the discrepancy to experimental differences.

Garg et al. (2023) logged their batch of 256 TD errors once every 5,000 steps during training for 100,000 timesteps, producing $\approx$ 4000 samples which were aggregated. They also computed bootstrapped targets with online parameters. In contrast, we sample 10,000 TD errors with bootstrapped targets computed from target parameters at a single timestep instance, and we do not aggregate samples across timesteps.

## E.2 Continuous Control Benchmarks and Evaluation

As mentioned in Section 6.2, the evaluation metric in continuous control was the normalized IQM with 95% stratified bootstrap confidence intervals from Agarwal et al. (2021). Returns were normalized by a minimum value computed from the mean of 100 rollouts sampled from a uniform policy and the maximum possible return from the environment. When the maximum value was not specified, we used the maximum value of any single rollout attained by any of the baselines.

We benchmarked on four continuous control suites: DeepMind Control (Tassa et al., 2018; Tunyasuvunakool et al., 2020), MuJoCo (Todorov et al., 2012; Brockman et al., 2016), MetaWorld (Yu et al., 2020), and Box2D (Brockman et al., 2016). These environments were selected to be as extensive as possible. DeepMind Control and MetaWorld were chosen because of their diversity of tasks, while the MuJoCo and Box2D environments are popular benchmarks within the common interface of OpenAI Gym (Brockman et al., 2016), now Gymnasium (Farama Foundation, 2023). No citation exists for Gymnasium as of writing this paper, and we link to their GitHub repository `https://github.com/Farama-Foundation/Gymnasium` as suggested in `https://github.com/Farama-Foundation/Gymnasium/issues/82`.

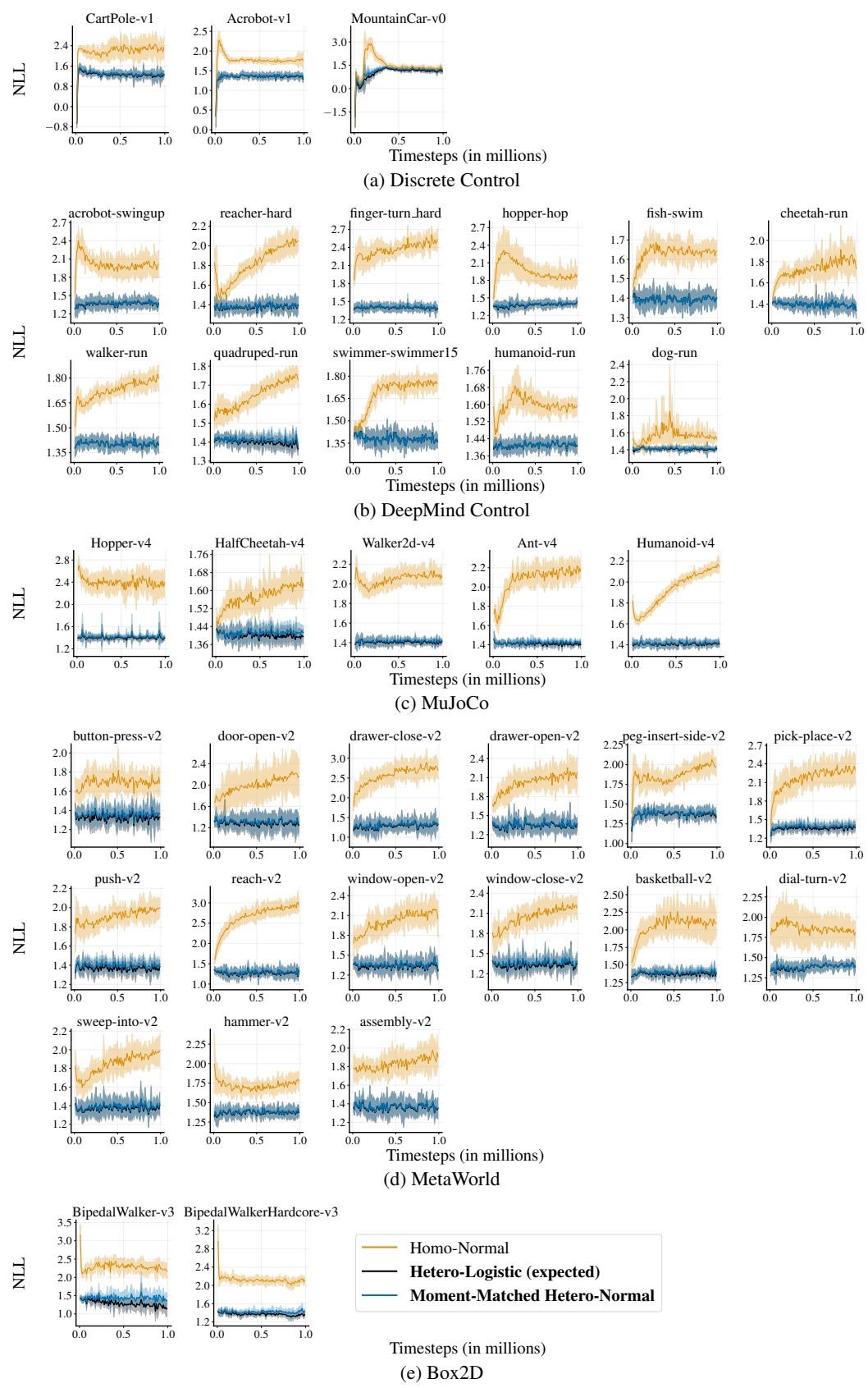

Figure 6: Negative Log-Likelihoods (NLLs) of the noise in Deep $Q$-Learning under different distributions throughout training (lower is better). Mean calculated per-task $\pm$ standard deviation. The legend for all graphs is in Figure 6e.

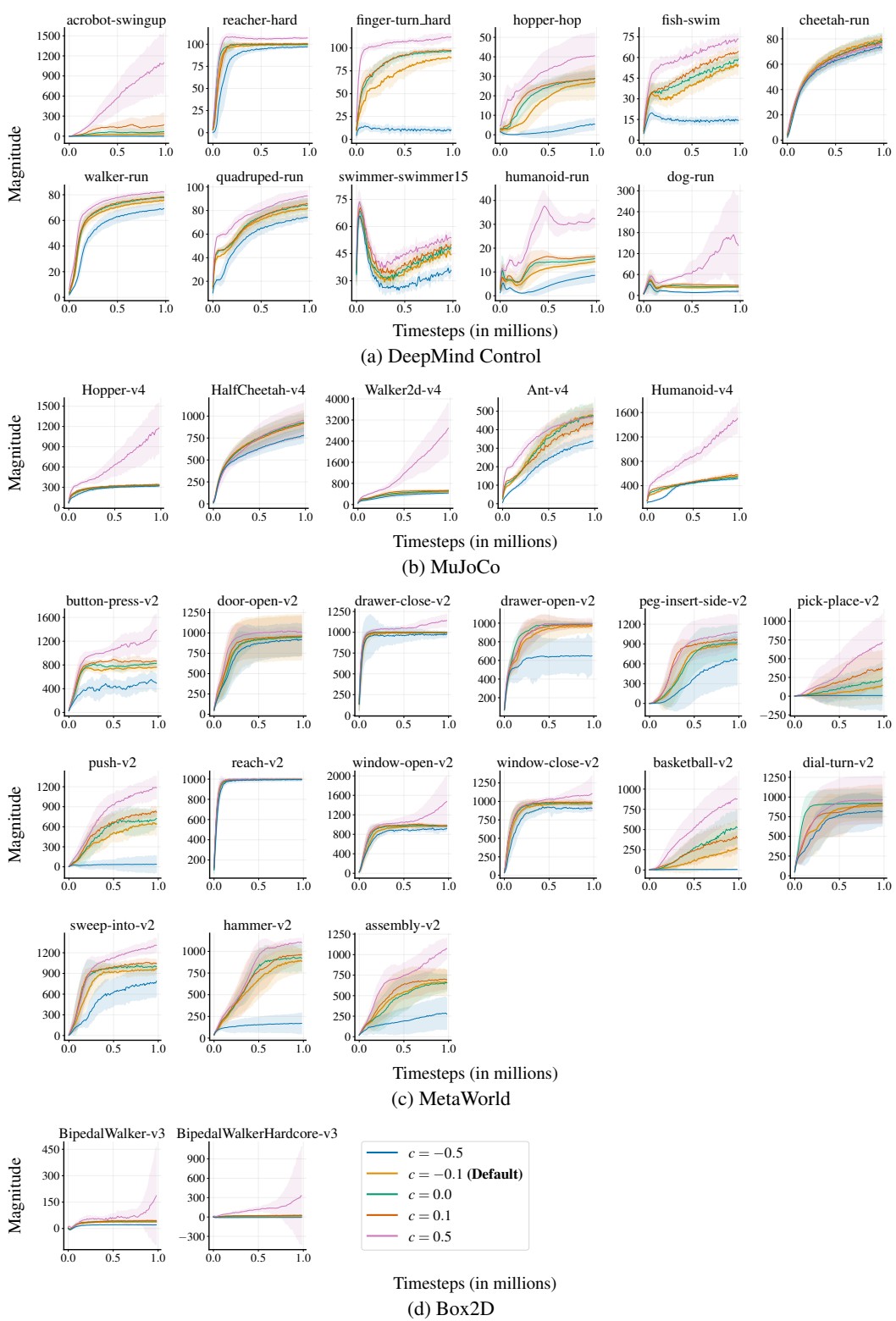

Figure 7: The effect of changing pessimism factor $c$ on the target $Q$-value in continuous control. IQM calculated per-task $\pm$ standard deviation. The legend for all graphs is in Figure 7d.

**DeepMind Control (DMC)** was designed to benchmark continuous control, over a broad range of agent morphologies. We selected agent morphologies that could be trained from states with a broad range of action spaces from 1 (`acrobot`) to 38 (`dog`). We did not benchmark on `humanoid_CMU` as this environment was not intended to be solved with RL from scratch, unlike the other baselines. The hardest task was selected from each of the agent morphologies. Properties of the 11 DMC tasks are presented in Table 1a.

**MetaWorld** was designed to have a diverse range of tasks to evaluate the generalization ability of learned policies. Each environment within MetaWorld is therefore made up of multiple tasks, all with the same underlying structure of an MDP but with different numerical values of their parameters. We follow the method of Seyde et al. (2022) to benchmark on a single MetaWorld task by first selecting an environment and then randomly selecting a set of numerical parameters. Each new instantiation of a MetaWorld task would result in a different set of hyperparameters. As such, we expect the error bars in the aggregate statistics of MetaWorld to be substantially larger than the other environments. We benchmark on tasks formed from the union of the ML1, MT10, and ML10 train tasks that a policy in MetaWorld would be trained on, as well as the five environments benchmarked in Seyde et al. (2022). Properties of the 15 MetaWorld tasks are presented in Table 1c.

**MuJoCo** was evaluated on the same tasks as SAC (Haarnoja et al., 2018b). These tasks were all locomotion-based. Properties of the 5 MuJoCo tasks are presented in Table 1b.

**Box2D** was evaluated on all continuous control tasks from states. Properties of the 2 Box2D tasks are presented in 1d.

### E.3 Discrete Control Baselines

Discrete control algorithms were implemented as described in Section 3.4. Hyperparameters used in discrete control algorithms are detailed in Tables 2 and 3. We provide explanations for these design choices as follows.

**DQN**: The original DQN algorithm in Mnih et al. (2015) was designed for pixel inputs. We modified DQN to use state inputs by using an architecture described in Section 3.4 we used in continuous control that was popular for use with state inputs. Conversely to the continuous control architecture, we found removing GroupNorm (Wu and He, 2018) was crucial to getting DQN to work. Similarly to the continuous control architecture, we found that changing the initialization and target network updating drastically improved performance. We also used the MSE and `Adam` (Kingma and Ba, 2014) optimizers as Ceron and Castro (2021) showed that this yielded improved performance over the Huber Loss (Huber, 1992) and RMSProp (Hinton et al., 2012) of the original DQN. Our implementation of DQN solves classic discrete control tasks that the CleanRL (Huang et al., 2022b) reproduction of the original DQN paper at `https://docs.cleanrl.dev/rl-algorithms/dqn/#experiment-results_1` could not solve.

**Dueling Double DQN (Dueling DDQN)** was a baseline modified from Hessel et al. (2018) designed to be as compatible with DoubleGum as possible. Rainbow evaluated six innovations to DQN: Double DQN (Van Hasselt et al., 2016), Dueling DQN (Wang et al., 2016), noisy networks (Fortunato et al., 2017), $n$-step returns, C51 distributional RL (Bellemare et al., 2017), and prioritized replay (Schaul et al., 2016). We only used the first two of these six innovations in DoubleGum. We did not find $n$-step returns effective in discrete domains we considered, nor prioritized replay. Distributional RL was incompatible with DoubleGum, while Schwarzer et al. (2023) did not find noisy networks advantageous.

DoubleDQN was implemented following Van Hasselt et al. (2016) by computing bootstrapped targets of $Q_\psi^{\text{new}}(s, \max_a Q_\theta^{\text{new}}(s, a))$. Dueling DQN was implemented following Wang et al. (2016), with the advantage and value heads having two layers with a hidden layer of size 256 and ReLU activations. The stability of Dueling DQN was greatly improved by setting the biases of both dueling heads to 0.

**DoubleGum** was implemented as Dueling DDQN with an additional variance head described in Section 3.4.

Table 1: Properties of Continuous Control Environments

(a) DeepMind Control

| Environment | Task | Action Dimension | Maximum Return | Minimum Return |
|---|---|---|---|---|
| acrobot | swingup | 1 | 1000 | 3.252 |
| reacher | hard | 2 | 1000 | 8.547 |
| finger-turn | hard | 2 | 1000 | 67.78 |
| hopper | hop | 4 | 1000 | 0.07236 |
| fish | swim | 5 | 1000 | 70.99 |
| cheetah | run | 6 | 1000 | 3.647 |
| walker | run | 6 | 1000 | 22.96 |
| quadruped | run | 12 | 1000 | 108.2 |
| swimmer | swimmer15 | 14 | 1000 | 157 |
| humanoid | run | 21 | 1000 | 0.877 |
| dog | run | 38 | 1000 | 4.883 |

(b) MuJoCo

| Task | Action Dimension | Maximum Return | Minimum Return |
|---|---|---|---|
| Hopper-v4 | 3 | 3572 | 18.52 |
| HalfCheetah-v4 | 6 | 11960 | -283.4 |
| Walker2d-v4 | 6 | 5737 | 2.753 |
| Ant-v4 | 8 | 6683 | -60.06 |
| Humanoid-v4 | 17 | 6829 | 122.5 |

(c) MetaWorld

| Task | Action Dimension | Maximum Return | Minimum Return |
|---|---|---|---|
| button-press-v2 | 4 | 10000 | 187.5 |
| door-open-v2 | 4 | 10000 | 277.1 |
| drawer-close-v2 | 4 | 10000 | 842.5 |
| drawer-open-v2 | 4 | 10000 | 631.8 |
| peg-insert-side-v2 | 4 | 10000 | 8.083 |
| pick-place-v2 | 4 | 10000 | 5.449 |
| push-v2 | 4 | 10000 | 30.62 |
| reach-v2 | 4 | 10000 | 776.1 |
| window-open-v2 | 4 | 10000 | 230.3 |
| window-close-v2 | 4 | 10000 | 306.7 |
| basketball-v2 | 4 | 10000 | 10.2 |
| dial-turn-v2 | 4 | 10000 | 125.6 |
| sweep-into-v2 | 4 | 10000 | 63.41 |
| hammer-v2 | 4 | 10000 | 395.1 |
| assembly-v2 | 4 | 10000 | 226.3 |

(d) Box2D

| Task | Action Dimension | Maximum Return | Minimum Return |
|---|---|---|---|
| BipedalWalker-v3 | 4 | 300 | -99.97 |
| BipedalWalkerHardcore-v3 | 4 | 300 | -107.9 |

Table 2: Shared Hyperparameters of Benchmarked Algorithms

| Hyperparameter | Value |
|---|---|
| Evaluation Episodes | 10 |
| Evaluation Frequency | Maximum Timesteps / 100 |
| Discount Factor $\gamma$ | 0.99 |
| $n$-Step Returns | 1 step |
| Replay Ratio | 1 |
| Replay Buffer Size | 1,000,000 |
| Maximum Timesteps | 1,000,000 |

Table 3: Hyperparameters for Discrete Control

| Hyperparameter | Value |
|---|---|
| Starting Timesteps | 2,000 |
| Maximum Timesteps | 100,000 |
| Exploration | Policy Churn |
| Optimizer | `Adam` |
| Learning rate | 3e-4 |
| Number of groups in network GroupNorm | 0 |
| Network structure | `Linear(256), ReLU, Linear(256), ReLU` |

Table 4: Hyperparameters for Continuous Control

| Hyperparameter | Value |
|---|---|
| Starting Timesteps | 10,000 |
| Maximum Timesteps | 1,000,000 |
| Exploration Noise | 0.2 |
| Policy Noise in Critic Loss | 0.1 |
| Policy Noise in Actor Loss | 0.1 |
| Actor optimizer | `Adam` |
| Actor learning rate | 3e-4 |
| Critic optimizer | `Adam` |
| Critic learning rate | 3e-4 |
| Number of groups in Actor GroupNorm | 16 |
| Number of groups in Critic GroupNorm | 16 |
| Critic target networks EMA $\eta_\phi$ | 5e-3 |
| Actor target networks EMA | 1 |
| Critic structure | `Linear(256), GroupNorm, ReLU`
`Linear(256), GroupNorm, ReLU`
`Linear(256), GroupNorm, ReLU` |
| Actor structure | `Linear(256), GroupNorm, ReLU` |

Table 5: Pessimism Hyperparameters in Continuous Control

| Algorithm | Pessimism Hyperparameter | | | | |
|---|---|---|---|---|---|
| | Default | DeepMind Control | MuJoCo | MetaWorld | Box2D |
| **DoubleGum (ours)** | $-0.1$ | $-0.1$ | $-0.5$ | $0.1$ | $-0.1$ |
| DDPG/TD3 | Twin | Single | Twin | Single | Twin |
| SAC | Twin | Single | Twin | Single | Twin |
| XQL | Twin ($\beta = 5$) | Single (3) | Single (5) | Twin (2) | Twin (5) |
| QR-DDPG | Single | Single | Twin | Single | Twin |
| FinerTD3 | 1 | 1 | 3 | 3 | 1 |

### E.4 Continuous Control Baselines

Continuous control algorithms were implemented as described in Section 3.4. Hyperparameters used in continuous control algorithms are detailed in Tables 2 and 4. Pessimism hyperparameters are presented in Table 5 and were found following results in Appendix F.2.

As mentioned, all implementations used networks with two hidden layers of width 256, with orthogonal initialization (Saxe et al., 2013) and GroupNorm (Wu and He, 2018). Following Kostrikov (2021), target network parameters were updated with an EMA of $5e - 3$ in the critic and $0$ in the actor. All these design choices differ from their original implementations but improved aggregate performance. We provide explanations for these design choices as follows.

**DDPG** was introduced in Lillicrap et al. (2015) and Fujimoto et al. (2018) updated the design choices of DDPG to empirically improve its performance. In addition to the existing changes, our implementation uses the noise clipping scheme in the actor specified by Laskin et al. (2021).

**TD3** was implemented with three changes from Fujimoto et al. (2018). First, we update the actor once per critic update – ie using a delay of 1. This is such that the only hyperparameter change between our DDPG and TD3 is the use of Twin Networks. Secondly, we update the actor to maximize the mean of two critics rather than a single critic, a design choice we found empirically reduced variance between training runs. Thirdly, we do not compute the EMA of actor-network parameters. Removing this EMA improves sample efficiency but at the cost of higher variance.

**FinerTD3 (our introduced baseline)** was implemented with the same hyperparameters as TD3 but with an ensemble of 5 critic networks. We chose to use 5 networks because we tuned the pessimism factor hyperparameter of DoubleGum over 5 values. The 5 critics in FinerTD3 enable five values of pessimism to be used. Pessimism of FinerTD3 is adjusted in the bootstrapped targets. The 5 critic values are sorted by decreasing positivity, and the $i^{\text{th}}$ smallest value is used as the target critic value in the bootstrapped targets.

**SAC** was implemented with hyperparameters from Kostrikov (2021), which we found improved performance. Kostrikov (2021) differs from Haarnoja et al. (2018b) in two additional ways from those mentioned. The standard deviation in the actor was clipped to $[-10, 2]$, and the target entropy was the action dimension divided by 2 instead of just the action dimension.

**XQL** Garg et al. (2023) presents two off-policy algorithms: X-TD3 and X-SAC. We use X-TD3 to be consistent with the DDPG fixed-variance actor of DoubleGum and refer to it throughout as XQL. XQL tunes two hyperparameters per task: the use of twin networks/not and scalar hyperparameter $\beta$. We swept over the same $\beta$-values as Garg et al. (2023): 1, 2, 5 without Twin Critics and 3, 4, 10 and 20 with Twin Critics. $\beta$ was tuned in the same way as pessimism – we found a default $\beta$ value and a $\beta$ tuned per-suite. $\beta$ values are presented in Table 5 and were found following results in Appendix F.2.

**MoG-DDPG** is formed by combining a Mixture-of-Gaussians (MoG) critic with DDPG. The MoG critic was introduced in Appendix A of Barth-Maron et al. (2018) and improved by Shahriari et al. (2022). The latter paper combines the MoG critic with DDPG with distributed training, but we remove the distributed training component because we do not use it in DoubleGum.

**QR-DDPG (our introduced baseline)** combines the quantile regression method of Dabney et al. (2018b) with a DDPG actor. Although Ma et al. (2020); Wurman et al. (2022) and Teng et al. (2022) have combined quantile regression with SAC, we combine it with DDPG because DoubleGum is built on top of DDPG. Like Dabney et al. (2018b), we use 201 quantiles, but these are initialized with orthogonal initialization and are optimized with the MSE, rather than the Huber loss. QR was developed for discrete control and uses the Huber loss with the RMSProp optimizer popular in discrete control methods. We found better performance with the MSE and Adam optimizer, perhaps confirming the result of Ceron and Castro (2021) in distributional RL for continuous control.

**DoubleGum** was implemented as DDPG with a variance head described in Section 3.4.

### E.5 Compute Requirements

A single training run for discrete control may take up to 3 to 5 minutes on a laptop with an Intel Core i9 CPU, NVIDIA 1050 GPU and 31.0 GiB of RAM. On the same system, a single training run for continuous control takes 1 - 2 hours.

Table 6: Discrete Control Numerical Results

| Task | Score at 100K timesteps (IQM over 12 seeds) | | |
|---|---|---|---|
| | DoubleGum (ours) | DQN | DuelingDDQN |
| CartPole-v1 | $500 \pm 113.4$ | $475 \pm 105.5$ | $496.9 \pm 89.1$ |
| Acrobot-v1 | $-62.78 \pm 1.775$ | $-73.52 \pm 5.191$ | $-64.12 \pm 17.15$ |
| MountainCar-v0 | $-98.17 \pm 2.45$ | $-99.37 \pm 5.914$ | $-98.75 \pm 30.73$ |

The overwhelming majority of our experiments were run on private infrastructure. This cluster had a mixture of Intel Broadwell, Skylake, Cascade Lake, AMD Rome, AMD Milan CPUs, and NVIDIA P100s, V100s, and A100s GPUs. Benchmarking continuous control took roughly ten times longer than benchmarking discrete control. Multi-threaded experiments for continuous control running twelve seeds in parallel took 5 - 8 hours. 8 algorithms (DoubleGum, DDPG, TD3, MoG-Critics, SAC, XQL, QR-DDPG, FinerTD3) were benchmarked over 33 continuous control environments, and there were further runs for hyperparameter sweeps (4 for DoubleGum, 1 for SAC, 6 for XQL, 1 for QR-DDPG and 4 for FinerTD3), yielding 24 runs in total. These algorithms were run at least 10 times for development and hyperparameter tuning. This yields a lower bound of $8 \times 33 \times 24 \times 10 = 63360$ hours (7.23 years) of computation.

Assuming that all experiments were run on Tesla V100-SXM2-16GB (TDP of 250W), the cluster it was run on had a carbon efficiency of 0.0006 kgCO$_2$eq/kWh (that of the surrounding power grid) and that there were 63360 hours of cumulative computation, the total emissions were 9.51 kgCO$_2$eq, equivalent to driving 36km in an average car. Estimations were conducted using the MachineLearning Impact calculator presented in Lacoste et al. (2019).

# F Further Results

## F.1 Adjusting the Pessimism of DoubleGum

Figure 8 shows that sample efficiency is sensitive to the pessimism factor $c$ adjusting pessimism per suite greatly impacts sample efficiency. The best performing $c$ was $c = -0.1$, and was thus set as the default pessimism factor value.

Figure 9 shows that the performance of DoubleGum may be improved when the degree of pessimism is changed per suite. This graph was used to determine what pessimism factor to use in each suite, whose values are reported in Table 5.

## F.2 Adjusting the Pessimism of Baseline Algorithms

This section presents graphs used to determine which pessimism values to use for baseline algorithms. All final values are reported in Table 5.

Figure 10 shows that sample efficiency is sensitive to the use of pessimism determined by the use of Twin Networks/not. In aggregate, each method was improved by using Twin Networks. Twin networks were therefore set as the default pessimism option for all baseline algorithms apart from QR-DDPG, because Twin Networks was not used with quantile regression in (Dabney et al., 2018b). Figure 11 was used to determine whether to use Twin Networks/not on a per suite basis.

Similarly, Figures 14 and 15 were respectively used to determine pessimism hyperparameters for FinerTD3. In these two graphs, numbers refer to the $i^{\text{th}}$ smallest value returned by the ensemble of target critics. Finally, Figures 12 and 13 were respectively used to determine pessimism and $\beta$ hyperparameters for XQL.

## F.3 Discrete Control

Table 6 presents results for discrete control at 100K timesteps.

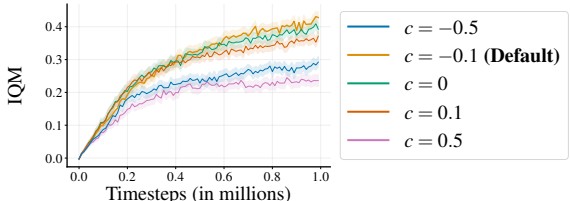

Figure 8: Adjusting the pessimism factor $c$ in DoubleGum, IQM normalized score over 33 tasks in 4 suites with 95% stratified bootstrap CIs.

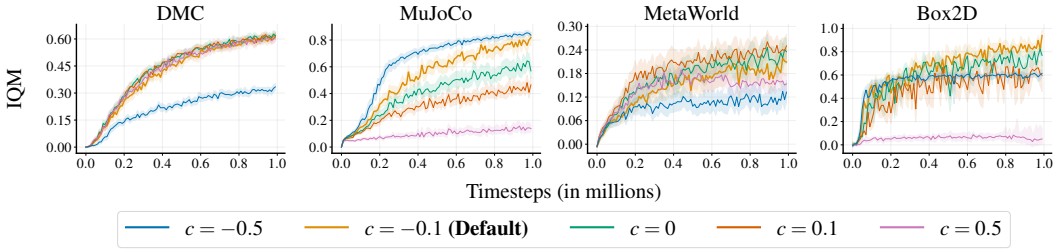

Figure 9: Adjusting pessimism in DoubleGum, per-suite IQM with 95% stratified bootstrap CIs.

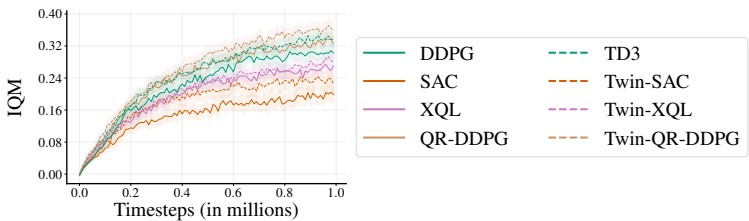

Figure 10: Adjusting pessimism of baseline algorithms with the use of Twin Networks/not, IQM normalized score over 33 tasks in 4 suites with 95% stratified bootstrap CIs. Methods that default to use Twin Networks are dashed.

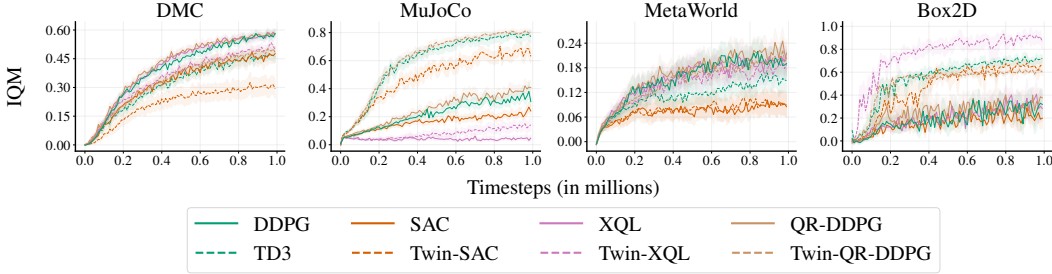

Figure 11: Adjusting pessimism of baseline algorithms with the use of Twin Networks/not, per-suite IQM normalized score with 95% stratified bootstrap CIs. Methods that default to use Twin Networks are dashed.

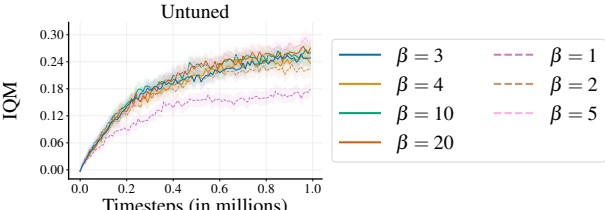

Figure 12: Adjusting pessimism of XQL, IQM normalized score over 33 tasks in 4 suites with 95% stratified bootstrap CIs. Methods that use Twin Networks are dashed.

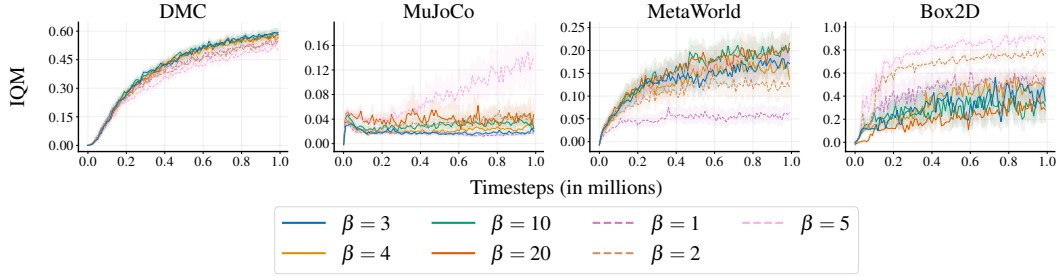

Figure 13: Adjusting pessimism of XQL, per-suite IQM normalized score with 95% stratified bootstrap CIs. Methods that use Twin Networks are dashed.

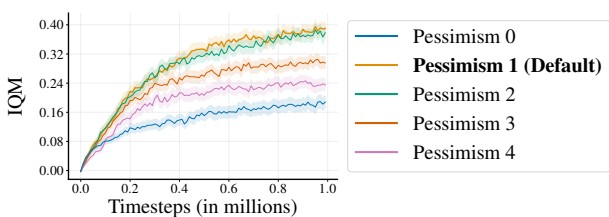

Figure 14: Adjusting pessimism of FinerTD3, IQM normalized score over 33 tasks in 4 suites with 95% stratified bootstrap CIs.

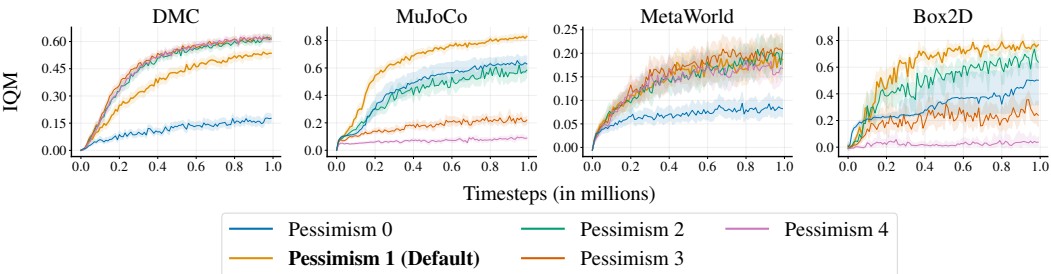

Figure 15: Adjusting pessimism of FinerTD3, per-suite IQM normalized score with 95% stratified bootstrap CIs.

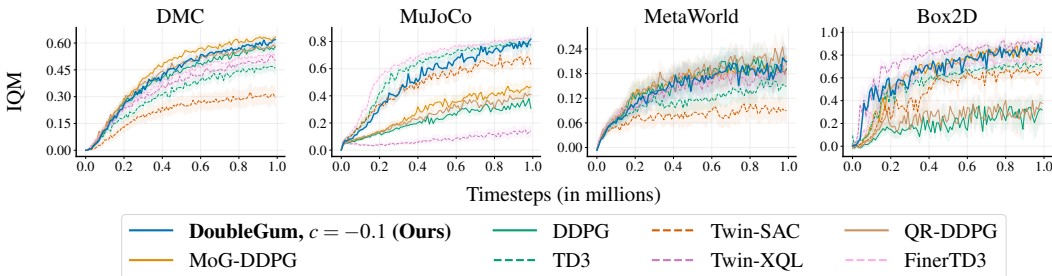

Figure 16: Continuous control with default parameters, per-suite IQM normalized score with 95% stratified bootstrap CIs. Methods that default to use Twin Networks are dashed.

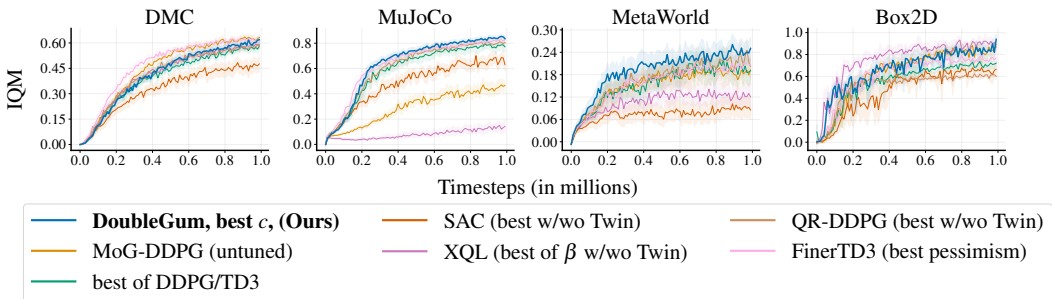

Figure 17: Continuous control with the best pessimism hyperparameters tuned per suite, per-suite IQM normalized score with 95% stratified bootstrap CIs.

## F.4 Continuous Control with Default Pessimism

Figures 16 and 18 respectively present aggregate per-suite and per-task results of DoubleGum benchmarked against baseline algorithms with default pessimism values.

Table 7 presents the performance of all algorithms with default pessimism at 1 million timesteps. This graph has six subsections. The first four subsections present per-task results from DeepMind Control, MuJoCo, MetaWorld, and Box2D, respectively, corresponding to results from 18. The next subsection presents per-suite aggregate results, corresponding to Figure 16, while the last subsection presents aggregate results over all tasks and suites, corresponding to Figure 1. In aggregate results, only the IQM is reported.

## F.5 Continuous Control with Pessimism adjusted Per-Suite

Figures 17 and 19 respectively present aggregate per-suite and per-task results of DoubleGum benchmarked against baseline algorithms with the best pessimism values adjusted per-suite.

Table 8 presents the performance of all algorithms with default pessimism at 1 million timesteps. This graph has six subsections. The first four subsections present per-task results from DeepMind Control, MuJoCo, MetaWorld, and Box2D, respectively, corresponding to results from 19. The next subsection presents per-suite aggregate results, corresponding to Figure 17, while the last subsection presents aggregate results over all tasks and suites, corresponding to Figure 1. In aggregate results, only the IQM is reported.

Table 7: Continuous control with default pessimism hyperparameters score after 1M timesteps.

| Suite/Task | Score at 1M timesteps (IQM ± standard deviation where appropriate) | | | | | | | |
| --- | --- | --- | --- | --- | --- | --- | --- | --- |
| | **DoubleGum** | MoG-DDPG | DDPG | TD3 | Twin-SAC | Twin-XQL | QR-DDPG | FinerTD3 |
| acrobot-swingup | 330.2 ± 79.44 | **364.9 ± 71.75** | 334.7 ± 96.92 | 7.315 ± 26.53 | 6.568 ± 6.164 | 38.84 ± 91.41 | 356.7 ± 88.26 | 8.114 ± 15.96 |
| reacher-hard | **979.4 ± 54.28** | 976.9 ± 4.985 | 975.7 ± 26.78 | 975.9 ± 31.13 | 976.2 ± 27.21 | 975.1 ± 4.506 | 972.9 ± 14.86 | 972.4 ± 25.15 |
| finger-turn_hard | 931.8 ± 98.91 | 935.4 ± 77.4 | 909.1 ± 50.68 | 969.7 ± 28.51 | 951.5 ± 63.21 | 967.2 ± 59.48 | 924.3 ± 73.62 | **974.1 ± 24.9** |
| hopper-hop | 306.7 ± 101 | **313.3 ± 99.74** | 305.7 ± 80.12 | 123.4 ± 69.37 | 0.007008 ± 0.03587 | 135.9 ± 65.21 | 304.3 ± 69.23 | 116 ± 51.71 |
| fish-swim | 675.7 ± 58.5 | **758.1 ± 62.68** | 710.2 ± 94.54 | 517.9 ± 111.2 | 344.9 ± 251.5 | 632.7 ± 99.64 | 695.2 ± 54.08 | 675 ± 118.8 |
| cheetah-run | **883.1 ± 22.51** | 844.9 ± 53.54 | 804.5 ± 64.31 | 745 ± 52.78 | 708.8 ± 44.35 | 761.3 ± 56.24 | 785.9 ± 63.43 | 741.7 ± 43.59 |
| walker-run | **783.6 ± 26.97** | 778.7 ± 21.86 | 755.6 ± 28.29 | 696 ± 39.53 | 538.6 ± 311.1 | 705.5 ± 114.6 | 743.9 ± 29.61 | 731.4 ± 109.2 |
| quadruped-run | **835.1 ± 65.11** | 818.2 ± 76.37 | 736 ± 87.58 | 677.2 ± 169.6 | 677 ± 181.7 | 743.9 ± 100.6 | 772.4 ± 80.69 | 739.8 ± 117.1 |
| swimmer-swimmer15 | 608.9 ± 137.8 | **623.4 ± 81.98** | 531.5 ± 104.1 | 477.8 ± 139.2 | 225.4 ± 130.5 | 435.5 ± 148.5 | 499.8 ± 64.12 | 612.5 ± 82.18 |
| humanoid-run | **142.5 ± 12.6** | 96.97 ± 12.58 | 119.6 ± 25.09 | 39.51 ± 62.78 | 0.8364 ± 0.234 | 119.2 ± 62.17 | 117.8 ± 17.68 | 1.452 ± 47.32 |
| dog-run | 187.1 ± 13.14 | 138.8 ± 20.58 | 177.4 ± 12.17 | 209.2 ± 25.22 | 5.226 ± 0.7772 | 229.5 ± 20.56 | 155.5 ± 22.55 | **234.4 ± 29.06** |
| Hopper-v4 | 1399 ± 668.1 | 1211 ± 319.2 | 1348 ± 740.2 | **2589 ± 939.2** | 942 ± 135.3 | 94.89 ± 122.7 | 1641 ± 554.7 | 2183 ± 890.6 |
| HalfCheetah-v4 | **10710 ± 608.3** | 9398 ± 1044 | 9552 ± 1557 | 10020 ± 1390 | 7171 ± 1000 | 9694 ± 1571 | 9148 ± 1476 | 10420 ± 328.8 |
| Walker2d-v4 | 4148 ± 1439 | 2196 ± 1127 | 1466 ± 470.3 | 3868 ± 593.1 | 2894 ± 1109 | 152.7 ± 79.47 | 2205 ± 890.8 | **4276 ± 357.1** |
| Ant-v4 | 6046 ± 552.5 | 4644 ± 966.2 | 976.9 ± 302.6 | 5645 ± 914 | 5908 ± 811.3 | 3652 ± 1685 | 2245 ± 761.4 | **6048 ± 464.4** |
| Humanoid-v4 | 5645 ± 904.1 | 1715 ± 783.4 | 2023 ± 604.9 | 5241 ± 302.4 | 5286 ± 703.2 | 160.2 ± 32.99 | 1814 ± 465.6 | **5668 ± 339.5** |
| button-press-v2 | **1436 ± 1241** | 919.6 ± 1066 | 1093 ± 1016 | 635.5 ± 566 | 1366 ± 1133 | 1128 ± 886.1 | 1424 ± 898.3 | 835 ± 1190 |
| door-open-v2 | 3671 ± 1606 | **4288 ± 1431** | 2691 ± 1415 | 3784 ± 809.1 | 4232 ± 454.2 | 3956 ± 941 | 3818 ± 1258 | 4083 ± 1059 |
| drawer-close-v2 | **4839 ± 1726** | 4706 ± 1578 | 3880 ± 1916 | 4743 ± 1718 | 3039 ± 2206 | 4750 ± 1560 | 4178 ± 1788 | 4808 ± 1324 |
| drawer-open-v2 | 2762 ± 1212 | **4074 ± 1421** | 2510 ± 1467 | 2951 ± 1511 | 2617 ± 1642 | 2256 ± 1819 | 2820 ± 1463 | 2048 ± 1187 |
| peg-insert-side-v2 | 1226 ± 1769 | 431.4 ± 1417 | 402.4 ± 1855 | 432 ± 1191 | 7.378 ± 2.552 | 317.8 ± 579.6 | 339.8 ± 1211 | **2146 ± 1752** |
| pick-place-v2 | 12.49 ± 110.5 | 212.5 ± 355 | **509.3 ± 672.7** | 7.501 ± 478.7 | 5.553 ± 2.761 | 16.98 ± 378 | 451.2 ± 847.2 | 6.942 ± 732.7 |
| push-v2 | 191.1 ± 1282 | 310.9 ± 1005 | **777.9 ± 928.9** | 42.5 ± 930.5 | 25.99 ± 76.78 | 415.2 ± 971.5 | 74.32 ± 847.7 | 729.7 ± 1571 |
| reach-v2 | 1746 ± 1127 | 2889 ± 909.7 | 2992 ± 1466 | 2092 ± 1304 | **3390 ± 1243** | 2552 ± 1066 | 3068 ± 1231 | 2735 ± 1113 |
| window-open-v2 | 2668 ± 1515 | 2501 ± 1764 | 3029 ± 1476 | 3452 ± 1150 | 491.3 ± 1326 | **3916 ± 1023** | 2294 ± 1551 | 2786 ± 1488 |
| window-close-v2 | 4404 ± 712 | 4520 ± 214.5 | 4352 ± 342.3 | 4202 ± 1145 | 4372 ± 704.6 | **4574 ± 866.3** | 4022 ± 1191 | 4291 ± 556 |
| basketball-v2 | 780.9 ± 1238 | 602.4 ± 1106 | 809.1 ± 901.4 | 627.9 ± 648.4 | 9.246 ± 197.2 | 1126 ± 1670 | **1972 ± 1197** | 524.4 ± 702.6 |
| dial-turn-v2 | 1214 ± 952.6 | 1552 ± 1250 | **2385 ± 1486** | 1228 ± 1206 | 2192 ± 1593 | 1446 ± 1020 | 2157 ± 1465 | 1258 ± 1034 |
| sweep-into-v2 | **4237 ± 1348** | 1417 ± 1448 | 3007 ± 1677 | 1977 ± 1907 | 530.2 ± 1712 | 3650 ± 1591 | 2203 ± 2026 | 1496 ± 2042 |
| hammer-v2 | **4308 ± 1215** | 3500 ± 1344 | 2664 ± 1477 | 1340 ± 1099 | 487.5 ± 945 | 2812 ± 1869 | 3620 ± 1311 | 2866 ± 1886 |
| assembly-v2 | **2606 ± 660.4** | 1941 ± 1241 | 2265 ± 861.8 | 2085 ± 801.6 | 947.2 ± 983.3 | 439.6 ± 1429 | 2376 ± 1032 | 1528 ± 921.1 |
| BipedalWalker-v3 | 315.2 ± 12.26 | 325.4 ± 8.892 | 218.6 ± 115.9 | 321.2 ± 2.343 | 321.1 ± 21.67 | **331.9 ± 6.717** | 247.4 ± 72.5 | 322.1 ± 6.48 |
| BipedalWalkerHardcore-v3 | **189 ± 86.95** | 152.6 ± 52.39 | -71.08 ± 30.12 | 18.27 ± 45.01 | -32.19 ± 56.1 | 114.1 ± 65.9 | -84.9 ± 17.95 | 53.3 ± 40.58 |
| DeepMind Control Aggregate | 0.6191 | **0.6326** | 0.5823 | 0.4588 | 0.293 | 0.5003 | 0.5787 | 0.5352 |
| MuJoCo Aggregate | 0.818 | 0.4663 | 0.3077 | 0.7716 | 0.6304 | 0.1423 | 0.4079 | **0.8305** |
| MetaWorld Aggregate | **0.209** | 0.193 | 0.1904 | 0.149 | 0.09027 | 0.1881 | 0.2017 | 0.1867 |
| Box2D Aggregate | **0.9391** | 0.8911 | 0.319 | 0.7204 | 0.6658 | **0.8659** | 0.3764 | 0.7679 |
| All Aggregate | **0.428** | 0.3709 | 0.3026 | 0.3358 | 0.226 | 0.2802 | 0.3255 | 0.39 |

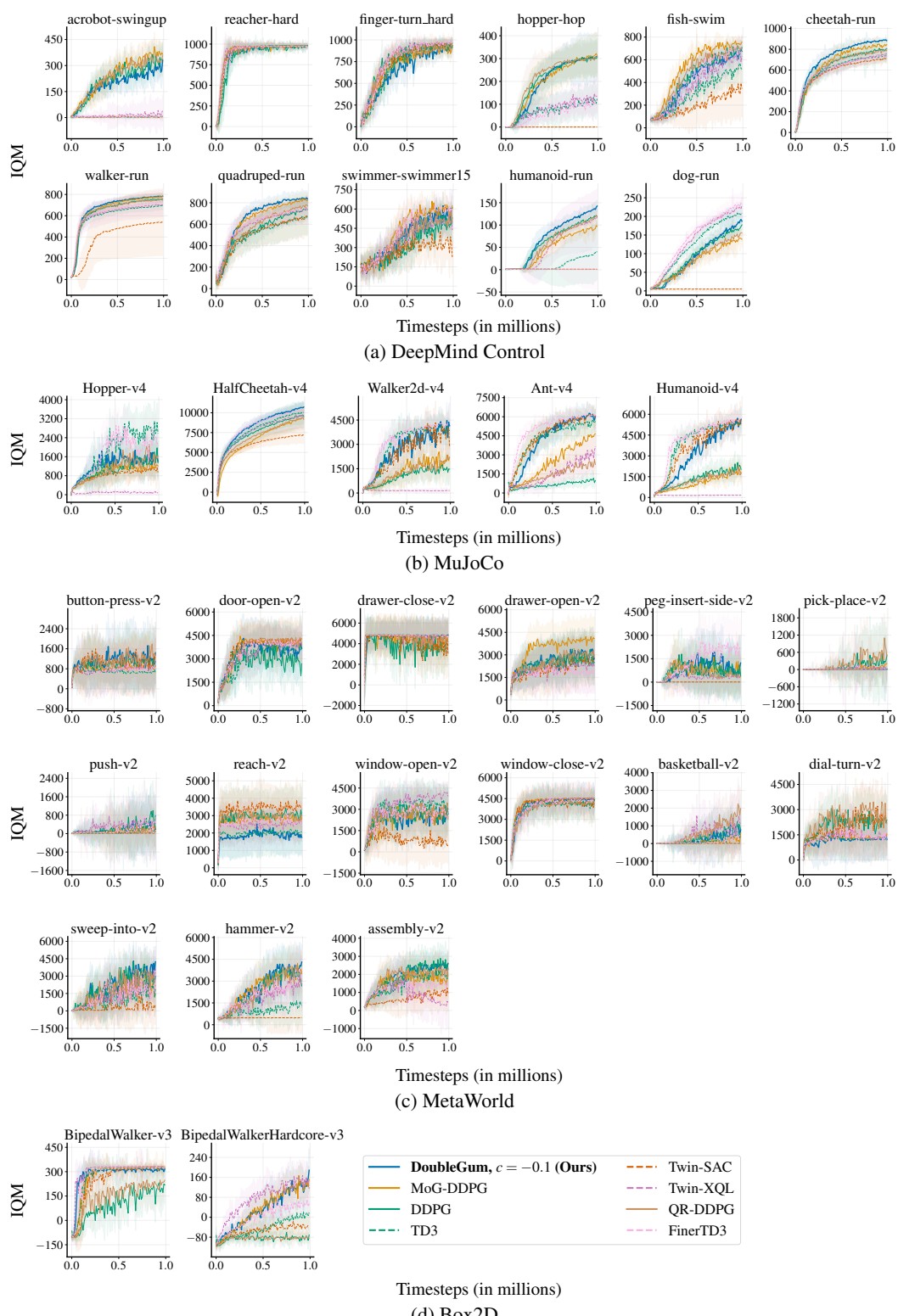

Figure 18: Continuous control with default pessimism hyperparameters, per-task IQM ± standard deviation. Methods that default to use Twin Networks are dashed. The legend for all graphs is in Figure 18d.

Table 8: Continuous control with the best pessimism hyperparameters adjusted per suite after 1M timesteps

| Suite/Task | Score at 1M timesteps (IQM ± standard deviation where appropriate) with best pessimism | | | | | | |
|---|---|---|---|---|---|---|---|
| | **DoubleGum** | MoG-DDPG | DDPG/TD3 | SAC | XQL | QR-DDPG | FinerTD3 |
| acrobot-swingup | 330.2 ± 79.44 | **364.9 ± 71.75** | 334.7 ± 96.92 | 226.6 ± 134.5 | 356.7 ± 88.26 | 225.7 ± 131.1 | 283.3 ± 89.79 |
| reacher-hard | **979.4 ± 54.28** | 976.9 ± 4.985 | 975.7 ± 26.78 | 976.2 ± 56.58 | 972.9 ± 14.86 | 975.4 ± 13.21 | 976.3 ± 27.78 |
| finger-turn_hard | 931.8 ± 98.91 | 935.4 ± 77.4 | 909.1 ± 50.68 | **972.4 ± 25.53** | 924.3 ± 73.62 | 952.6 ± 65.92 | 956.4 ± 74.46 |
| hopper-hop | 306.7 ± 101 | 313.3 ± 99.74 | 305.7 ± 80.12 | 0.007651 ± 56.18 | 304.3 ± 69.23 | 346.8 ± 105 | **519.5 ± 47.02** |
| fish-swim | 675.7 ± 58.5 | 758.1 ± 62.68 | 710.2 ± 94.54 | 616.4 ± 226.4 | 695.2 ± 54.08 | 708.8 ± 38.73 | **761.6 ± 41.15** |
| cheetah-run | **883.1 ± 22.51** | 844.9 ± 53.54 | 804.5 ± 64.31 | 764.5 ± 74.55 | 785.9 ± 63.43 | 791.1 ± 69.23 | 699 ± 84.17 |
| walker-run | **783.6 ± 26.97** | 778.7 ± 21.86 | 755.6 ± 28.29 | 732 ± 199.4 | 743.9 ± 29.61 | 759.8 ± 24.46 | 777 ± 12.7 |
| quadruped-run | **835.1 ± 65.11** | 818.2 ± 76.37 | 736 ± 87.58 | 774 ± 62.96 | 772.4 ± 80.69 | 775 ± 54.76 | 724.4 ± 92.74 |
| swimmer-swimmer15 | 608.9 ± 137.8 | **623.4 ± 81.98** | 531.5 ± 104.1 | 437.8 ± 192.5 | 499.8 ± 64.12 | 552.7 ± 140.2 | 560.2 ± 106.4 |
| humanoid-run | **142.5 ± 12.6** | 96.97 ± 12.58 | 119.6 ± 25.09 | 0.9647 ± 0.1496 | 117.8 ± 17.68 | 122.4 ± 9.644 | 85.08 ± 6.825 |
| dog-run | **187.1 ± 13.14** | 138.8 ± 20.58 | 177.4 ± 12.17 | 22.63 ± 56.75 | 155.5 ± 22.55 | 145.3 ± 28.37 | 87.51 ± 32.47 |
| Hopper-v4 | **3290 ± 829.1** | 1211 ± 319.2 | 2589 ± 939.2 | 942 ± 135.3 | 2398 ± 946.4 | 94.89 ± 122.7 | 2183 ± 890.6 |
| HalfCheetah-v4 | 9874 ± 997.4 | 9398 ± 1044 | 10020 ± 1390 | 7171 ± 1000 | 9855 ± 1323 | 9694 ± 1571 | **10420 ± 328.8** |
| Walker2d-v4 | **4871 ± 403.7** | 2196 ± 1127 | 3868 ± 593.1 | 2894 ± 1109 | 4027 ± 962 | 152.7 ± 79.47 | 4276 ± 357.1 |
| Ant-v4 | 5681 ± 416.8 | 4644 ± 966.2 | 5645 ± 914 | 5908 ± 811.3 | **6185 ± 168.3** | 3652 ± 1685 | 6048 ± 464.4 |
| Humanoid-v4 | 5565 ± 160.7 | 1715 ± 783.4 | 5241 ± 302.4 | 5286 ± 703.2 | 5452 ± 243.1 | 160.2 ± 32.99 | **5668 ± 339.5** |
| button-press-v2 | 1436 ± 1241 | 919.6 ± 1066 | 1093 ± 1016 | 645.5 ± 728.3 | 1424 ± 898.3 | **1506 ± 1047** | 1086 ± 758.2 |
| door-open-v2 | 3671 ± 1606 | **4288 ± 1431** | 2691 ± 1415 | 3114 ± 1166 | 3818 ± 1258 | 3512 ± 1387 | 4268 ± 953.8 |
| drawer-close-v2 | **4839 ± 1726** | 4706 ± 1578 | 3880 ± 1916 | 4749 ± 1339 | 4178 ± 1788 | 4588 ± 2042 | 3741 ± 2037 |
| drawer-open-v2 | 2762 ± 1212 | **4074 ± 1421** | 2510 ± 1467 | 1710 ± 1514 | 2820 ± 1463 | 2972 ± 1382 | 1608 ± 1333 |
| peg-insert-side-v2 | **1226 ± 1769** | 431.4 ± 1417 | 402.4 ± 1855 | 9.693 ± 232.9 | 339.8 ± 1211 | 100.8 ± 1462 | 1092 ± 1699 |
| pick-place-v2 | 12.49 ± 110.5 | 212.5 ± 355 | **509.3 ± 672.7** | 4.821 ± 1.873 | 451.2 ± 847.2 | 26.68 ± 47.73 | 501.4 ± 963.9 |
| push-v2 | 191.1 ± 1282 | 310.9 ± 1005 | **777.9 ± 928.9** | 90.81 ± 745 | 74.32 ± 847.7 | 76.38 ± 95.72 | 256.8 ± 453.4 |
| reach-v2 | 1746 ± 1127 | 2889 ± 909.7 | 2992 ± 1466 | 2974 ± 1393 | **3068 ± 1231** | 2583 ± 1345 | 2402 ± 922.9 |
| window-open-v2 | 2668 ± 1515 | 2501 ± 1764 | **3029 ± 1476** | 1957 ± 1832 | 2294 ± 1551 | 1672 ± 1813 | 2964 ± 1365 |
| window-close-v2 | 4404 ± 712 | 4520 ± 214.5 | 4352 ± 342.3 | 4000 ± 1045 | 4022 ± 1191 | **4528 ± 232.3** | 4272 ± 260 |
| basketball-v2 | 780.9 ± 1238 | 602.4 ± 1106 | 809.1 ± 901.4 | 9.932 ± 11.24 | **1972 ± 1197** | 99.83 ± 745.6 | 1868 ± 959.1 |
| dial-turn-v2 | 1214 ± 952.6 | 1552 ± 1250 | **2385 ± 1486** | 1472 ± 1178 | 2157 ± 1465 | 1690 ± 1408 | 1101 ± 1255 |
| sweep-into-v2 | **4237 ± 1348** | 1417 ± 1448 | 3007 ± 1677 | 247.9 ± 1718 | 2203 ± 2026 | 1008 ± 1782 | 2844 ± 1934 |
| hammer-v2 | **4308 ± 1215** | 3500 ± 1344 | 2664 ± 1477 | 1180 ± 1650 | 3620 ± 1311 | 1892 ± 1387 | 4151 ± 1115 |
| assembly-v2 | **2606 ± 660.4** | 1941 ± 1241 | 2265 ± 861.8 | 218.1 ± 492.3 | 2376 ± 1032 | 164 ± 401.5 | 2067 ± 1507 |
| BipedalWalker-v3 | 315.2 ± 12.26 | 325.4 ± 8.892 | 321.2 ± 2.343 | 321.1 ± 21.67 | 311.3 ± 18.27 | **331.9 ± 6.717** | 322.1 ± 6.48 |
| BipedalWalkerHardcore-v3 | **189 ± 86.95** | 152.6 ± 52.39 | 18.27 ± 45.01 | -32.19 ± 56.1 | -31.74 ± 18.9 | 114.1 ± 65.9 | 53.3 ± 40.58 |
| DMC Aggregate | 0.6191 | **0.6326** | 0.5823 | 0.4762 | 0.5918 | 0.5787 | 0.6121 |
| MuJoCo Aggregate | **0.8367** | 0.4663 | 0.7716 | 0.6304 | 0.1423 | 0.8061 | 0.8305 |
| MetaWorld Aggregate | **0.2505** | 0.193 | 0.1904 | 0.08515 | 0.1204 | 0.2017 | 0.2042 |
| Box2D Aggregate | **0.9391** | 0.8911 | 0.7204 | 0.6658 | 0.8659 | 0.601 | 0.7679 |
| All Aggregate | **0.4707** | 0.3709 | 0.3979 | 0.2719 | 0.2786 | 0.3924 | 0.4239 |

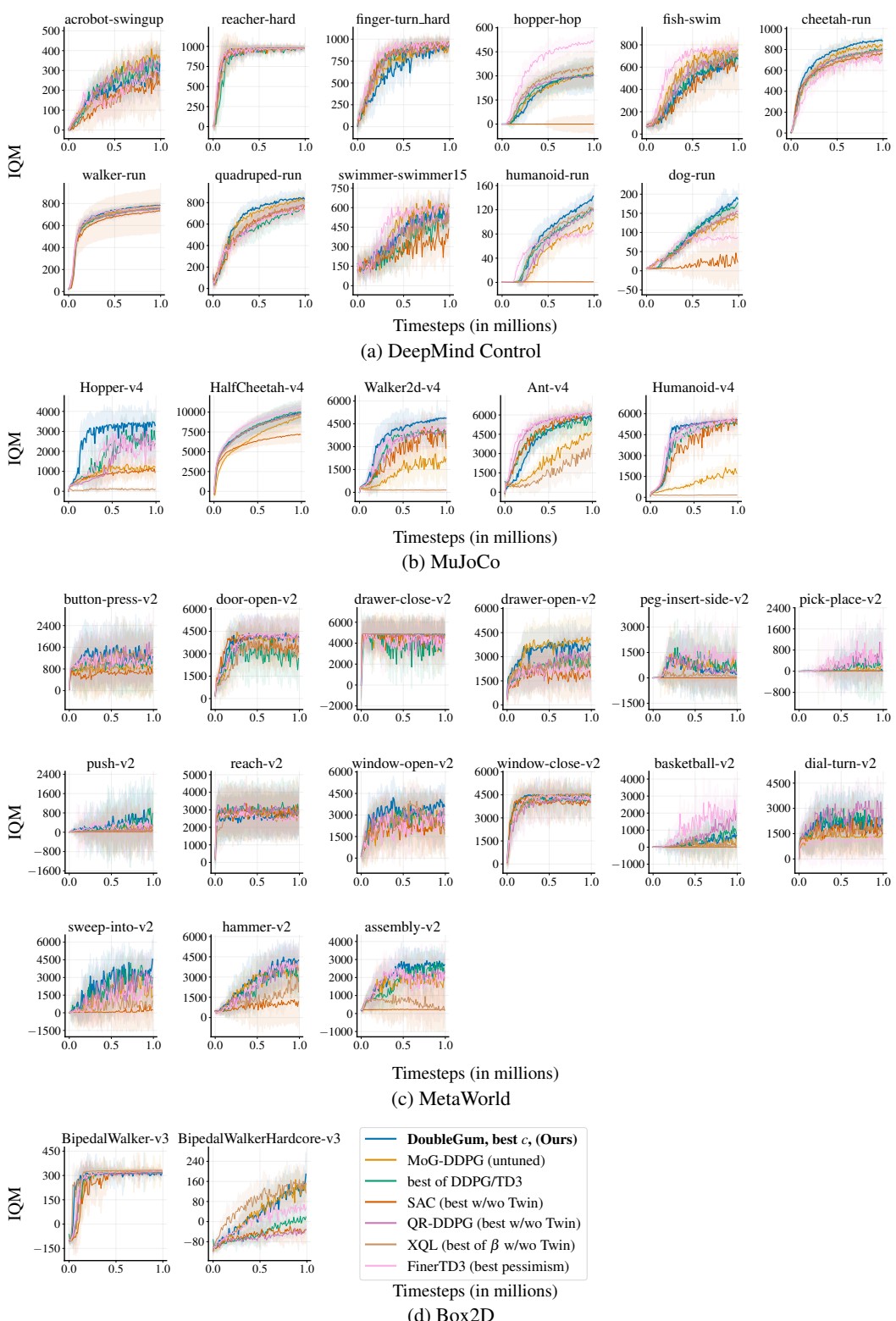

Figure 19: Continuous control with the best pessimism hyperparameters adjusted per suite, per-task IQM ± standard deviation. The legend for all graphs is in Figure 19d.

