# OpenReview forum: "Double Gumbel Q-Learning"
_NeurIPS.cc/2023/Conference — NeurIPS 2023 spotlight_

### Official Review · Reviewer_Mh9L · 2023-07-04

**Soundness:** 3 good
**Presentation:** 3 good
**Contribution:** 3 good
**Rating:** 6
**Confidence:** 2

**Summary:**

The authors propose a novel off-policy algorithm DoubleGum based on their new noise model which is claimed to be more stable for training and reaches higher performance for various discrete and continuous environments.

**Strengths:**

The paper is well-written and well organized. The authors proposed a novel noise model and off-policy DRL algorithm and analyse their strength through a set of extensive experiments. Some experimental results are interesting and strong. They also provide nice math supporting the paper.



**Weaknesses:**

Figures are a bit confusing. For example, it's a bit hard to see just from the figures that how stable is the new algorithm comparing to existing state-of-the-art. Also, I saw TD3, DDPG MoG-DDPG in figure3, then it is just DDPG (best w/wo Twin Networks) in figure 5. Makes me a bit confused and not sure whether DoubleGum with adjusting pessimism outperforms all the other algorithms or just DDPG?


**Questions:**

1.Curious why SAC is not included in the experiments. To my knowledge, it is a very strong off-policy algorithm. It is stable and outperforms DDPG and TD3 in a bunch of continuous tasks.
2.Can we claim DoubleGum with adjusting pessimism the new state-of-the art?
3.Will there be a way to measure how much more stable is DoubleGum(with fixed pessimism) across different tasks?Since in fixed pessimism, as shown in fig3, the performance seems to be sacrificed.


**Limitations:**

As authors mentioned in the paper, the proposed algorithm seems to be very sensitive to hyper-parameters, which is not very applicable if we need to tune pessimism per task.

---

> ### Author Rebuttal · Authors · 2023-08-09
>
> Many thanks for your review!
>
> We are happy to read that you found our algorithm novel and that our paper was well-written and organized.
>
> > TD3, DDPG MoG-DDPG in figure3, then it is just DDPG (best w/wo Twin Networks) in figure 5
>
> Figures 3 and 5 show two different ways of evaluating DoubleGum.  They correspond to two different methods in which we benchmark DoubleGum, namely 1) keeping hyperparameters constant (to their default values) across all tasks and suites 2) varying hyperparameters between suites, but constant within all tasks per suite.  We would like to point out that we do not vary hyperparameters across tasks belonging to the same suite, and thus we do not tune pessimism per task.  We realize that we did not make this clear in our original text, and, if accepted, would be happy to change it.
>
> The hyperparameter that we tune is pessimism.  For DoubleGum, this corresponds to changing pessimism factor $c$.  For DDPG, this corresponds to whether we use Twin Networks or not, resulting in our caption of "DDPG (best w/wo Twin Networks)".  As our implementations of DDPG with Twin Networks and TD3 are equivalent, we have updated the aforementioned caption to "Best of DDPG/TD3" in our 1-page `.pdf` whenever necessary.
>
> > whether DoubleGum with adjusting pessimism outperforms all the other algorithms or just DDPG?
>
> We have uploaded Figures 14 and 15 in our attached 1-page `.pdf`, which show aggregate learning curves across all 33 continuous control tasks.  In these graphs, DoubleGum outperforms all benchmark algorithms, including the newly benchmarked SAC and XQL.  More details are given in point 1 of our global rebuttal.
>
> > Curious why SAC is not included in the experiments
>
> We now benchmark against SAC and show that in Updated Figures 3 and 5, as well as Figures 14 and 15 in our attached 1-page `.pdf`, DoubleGum is empirically stronger.  More details are presented in Point 2 of our global rebuttal.
>
> > Can we claim DoubleGum with adjusting pessimism the new state-of-the art?
>
> We are hesitant to claim that DoubleGum is the new SOTA, due to the wide recent algorithmic innovations such as [1, 2], and the fact that most recent papers do not benchmark on as many tasks as we do, making comparisons difficult.  However, we would like to claim that DoubleGum is the best-performing core algorithm over all continuous control tasks in aggregate (Figures 14, 15), and would thus like to recommend that DoubleGum form the new core algorithm for many tasks, just as SAC was the core algorithm for [1], or TD3 was for [2].
>
> [1] P. D’Oro, M. Schwarzer, E. Nikishin, P.-L. Bacon, M. G. Bellemare, and A. Courville. Sample-
> efficient reinforcement learning by breaking the replay ratio barrier
>
> [2] S Fujimoto, WD Chang, EJ Smith, SS Gu, D Precup, D Meger. For SALE: State-Action Representation Learning for Deep Reinforcement Learning
>
> > a way to measure how much more stable is DoubleGum(with fixed pessimism) across different tasks?
>
> In our 1-page `.pdf`, Figures 14 and 15 show learning curves aggregated across all 33 continuous control tasks we benchmark on.  These learning curves show how consistently the algorithms perform across all tasks, and thus how stable the algorithms are between tasks.  DoubleGum achieves a higher learning curve than the baselines indicating that it is more stable across tasks than the baseline algorithms.
>
> >  the proposed algorithm seems to be very sensitive to hyper-parameters
>
> We would like to point out that while we have shown that DoubleGum is sensitive to the pessimism factor $c$, we do not think DoubleGum is any more sensitive to any other hyperparameters than our baselines are.
>
> > [DoubleGum] not very applicable if we need to tune pessimism per task.
>
> We would like to reiterate that in our evaluations, we do not tune per task and only per suite.  If accepted, we will update our text to make this clearer.  As we intend $c$ in DoubleGum to be similar to how the learning rate is used in stochastic gradient descent optimizers, we evaluate in two different ways: untuned with default parameters, and tuned per suite.  We go into more detail in Point 4 of our global rebuttal.
>
> > in fixed pessimism, as shown in fig3, the performance seems to be sacrificed
>
> We would expect the default parameter to have poorer performance than if we tuned it specifically for performance per task/suite.  Nevertheless, the aggregate learning curve for DoubleGum with fixed default pessimism still outperforms all baseline methods (Figure 14).

---

> > ### Comment · Reviewer_Mh9L · 2023-08-16
> > **Response to rebuttal**
> >
> > Thank the authors for the response to my questions and add additional experiments for SAC. I think I'll maintain my score.

---

### Official Review · Reviewer_je5E · 2023-07-06

**Soundness:** 2 fair
**Presentation:** 3 good
**Contribution:** 3 good
**Rating:** 6
**Confidence:** 4

**Summary:**

The paper provides empirical evidence that it is inaccurate to assume the temporal difference error follows a homoscedastic normal distribution. The paper proposes to use Gumbel distribution to make a replacement. The optimal action value is considered as the learned value estimation plus a noise sampled from a Gumbel distribution. The Bellman Equation has changed accordingly. Based on this idea, a new algorithm for both discrete and continuous control tasks is derived, called DoubleGum. The paper empirically tested DoubleGum in various tasks, including several relatively complex environments.

**Strengths:**

- The paper provides clear empirical evidence to explain that the temporal difference error is better fitted to a Guambel distribution rather than a normal distribution.

- The paper has a relatively complete related work list. The related work section considered other analyses of the noise in Q-learning and works proposing similar methods.

- DoubleGum empirically works on both discrete and continuous control tasks. To support this, in the experiment section, the paper empirically tested it on multiple environments, including both simple and complex tasks.

- Detailed hyperparameters and computational resources needed are reported, improving the reproducibility.



**Weaknesses:**

- The main concern comes from the gap between the theory and the practical algorithm. The whole idea is based on one assumption, which is, estimating the temporal difference error with a Gumbel distribution is better than Normal distribution. The Bellman equation is also modified according to the same intuition. However, when deriving the practical algorithm, the paper points out that the Q-function cannot be learned by Gumbel or logistic regression because there lacks a sufficient statistical estimator. So, DoubleGum instead used a generalized method of moments to match the logistic with a normal distribution.

- Although the empirical results did suggest DoubleGum learned faster than the baseline (DDPG), it is not clear to me if the advantage does come from the Gumbel distribution, as the practical algorithm is not using the Gumbel distribution as previous sections discuss.

- The other concern is mainly about the baseline choice. SAC is another algorithm that empirically works well for online learning. It estimates the action value with considering the entropy. The setting that uses a softmax instead of max is consistent with a part of the proposed method in the paper. It might be worth checking SAC’s performance in the tasks listed in paper, and comparing it with DoubleGum.


**Questions:**

I would appreciate it if the authors could explain the abovementioned concerns.

**Limitations:**

Yes

---

> ### Author Rebuttal · Authors · 2023-08-09
>
> Many thanks for your review!
>
> We are delighted to read your positive feedback on our experimental methodology and reproducibility, which we care deeply about.
>
> > The whole idea based on one assumption, estimating the temporal difference error with a Gumbel distribution is better than Normal distribution.
>
> We would like to point out that we estimate TD errors with *two* heteroscedastic Gumbel distributions (Equation 15), which we rewrite into a heteroscedastic Logistic (Equation 19).
>
> > DoubleGum instead used a generalized method of moments to match the logistic with a normal distribution.
>
> We would like to point out that treating the heteroscedastic Logistic distribution (Equation 19) as a *homo*scedastic Normal (Equation 5) is problematic, but generalized moment matching with a *hetero*scedastic Normal is not.  We realize that we did not make the distinction between hetero/homo clear enough in our paper, and will update the text if accepted.
>
> The generalized method of moments estimator has been widely used in the statistics and econometrics literature.  We were specifically inspired to use the generalized method of moments from Equations 42-45 of [1], which estimates the location and spread parameters of a Gumbel distribution in this manner.  The same method is presented in [2] to demonstrate sampling from a Gumbel distribution in the NumPy package.
>
> [1] Jowitt, P. W. (1979). The extreme-value type-1 distribution and the principle of maximum entropy.
>
> [2] Code sample at the bottom of the documentation for numpy.random.gumbel
>
> > not clear to me if the advantage does come from [modelling] the Gumbel distribution
>
> The advantage comes from modeling the Q-values heteroscedastically, which is motivated by our theory that reveals that the TD errors are heteroscedastic, not homoscedastic.  Learning a sample-dependent variance is not typically done in Q-learning, and not in our baselines, and therefore in our experiments with default hyperparameters (Updated Figure 3, Figure 14), is the source of our empirical advantage.
>
> > checking SAC’s performance in the tasks listed in paper, and comparing it with DoubleGum
>
> We now benchmark against SAC. and show that in Updated Figures 3 and 5, as well as Figures 14 and 15 in our attached 1-page `.pdf`, DoubleGum is empirically stronger.  More details are given in point 2 of our global rebuttal.
>
> We hope that our response helps bridge the gap between our mathematical analysis and the resulting algorithm, and shows stronger empirical evidence for the effectiveness of DoubleGum.

---

> > ### Comment · Reviewer_je5E · 2023-08-17
> > **Reply**
> >
> > Sorry for the late reply! I would like to thank the author for the detailed reply and attached experiment results. It addressed my concern, and I increased my score from 4 to 6.

---

### Official Review · Reviewer_uPm8 · 2023-07-06

**Soundness:** 2 fair
**Presentation:** 2 fair
**Contribution:** 2 fair
**Rating:** 4
**Confidence:** 4

**Summary:**

The paper proposes a combination of Gumbel noise with Q-learning. The idea is based on the high level observation that regular L-2 loss based Q-learning can be understood as maximum likelihood estimation with Gaussian noise. The paper derives the practical algorithm and shows some improvements over baselines.

**Strengths:**

The paper takes a relatively novel perspective on interpreting L-2 loss based Q-learning algorithm as maximum likelihood estimation, and derives an empirically novel algorithm based on the Gumbel noise in place of Gaussian noise. The resulting algorithm looks easy to implement and showcases some practical improvements.

**Weaknesses:**

Since the algorithm is heuristically motivated, the paper seems to lack a solid theoretical grounding as to whether the newly proposed algorithm would converge or not (not formally stated or proved in the paper). The empirical improvements are interesting but can use more results to further demonstrate the promise of this new approach.

**Questions:**

#### === ** Definition of Q-function and random variables ** ===

The definition of the equality in distribution look a bit confusing to me in Eqn 9,

$$Q_\theta(s,a) =_d Q*(s,a) - g_\theta(s,a)$$

Here $g_\theta(s,a)$ is a random variable whose distribution depends on $(s,a)$. As a result, the RHS is a random variable with a fixed probability law (once $\theta$ is fixed). However, the Q-function on the LHS is a single scalar at $(s,a)$. In general, how should we expect such an equality to hold? If we parameterize LHS by a neural network, it outputs a single scalar and the equality cannot hold in general.

Related to this, the equality of Eqn 10 seems to be an equality in scalar, i.e. both LHS and RHS are scalar quantities. However, $g_\theta(s,a)$ on the LHS is a random variable, and is in general not a scalar? The definition of $y*$ in Eqn 14, on the other hand, confirms that Eqn 10 should have been an equality in scalar since the RHS $y*(s,a)$ is a scalar.

#### === **Parameterization of Q-function** ===

Following the above discussion, in line 127 there is a use of distributional equality again, where LHS is $Q_\theta^\text{new}$. For this equality to hold, we do have to parameterize the Q-function output as a distribution at $(s,a)$ instead of a single scalar? A further clarification on this would be very helpful.

#### === **Convergence guarantee** ===

The algorithm proposed is very linked to max-entropy RL, the convergence property of which has been established. The L-2 based Q-learning algorithm is also convergent under suitable conditions. I wonder if there is a similar guarantee here, and if yes would be good to have in the main paper.

#### === **Empirical results** ===

Gumbel DDPG seems to improve over DDPG but not over TD3. I guess a main reason might be a TD3 uses double Q-learning, which is orthogonal to what's proposed here. Can we combine Gumbel trick with TD3 and show improvements over TD3? DDPG is in general a much more inferior baseline than TD3 due to the over-estimation bias, it'd be nice to showcase improvements over TD3 too.

From Fig 4 it seems that the performance is sensitive to the hyper-parameter $c$. How is the choice of $c=0$ selected in other experiments in comparison to alternative baselines? Is $c=0$ the best-performing hyper-parameter in general?



**Limitations:**

Discussed above.

---

> ### Author Rebuttal · Authors · 2023-08-10
>
> Many thanks for your review!
>
> We are very happy that you found both theoretical and empirical novelty within our paper, and that our resulting DoubleGum algorithm was practical and easy to implement.
>
> > The definition of the equality in distribution look a bit confusing to me in Eqn 9
>
> We have realized we should use an equality in samples $=$ whenever we've used a distributional equality $\overset{d}{=}$. Equation 9 constructs a new random variable $Q_\theta(s, a) \sim Q(s, \cdot)$ by defining a data-generating process that subtracts a random variable $g_{\theta, a}(s) \sim g(s)$ from the optimal $Q^\star(s, a)$ given an $\theta$ and $a$ index. As Equation 9 concerns samples and not random variables, it is not appropriate to use a distributional equality.
>
> $Q_\theta$s is a random variable because it varies throughout training. In supervised learning, it is standard to model the output of a neural network as a random variable -- in regression the neural network estimate is a random variable that differs from the optimal value by Normal noise. We work through the mathematics of this example in Appendix A.1. There should again be a sample equality in the equation between lines 472-3.
>
> > equality of Eqn 10 seems to be an equality in scalar i.e. both LHS and RHS are scalar quantities.
>
> The LHS of Eqn 10 involves samples from two random variables whose sum is constructed to be a constant, which is the RHS.
>
> > we do have to parameterize the Q-function output as a distribution at $(s, a)$ instead of a single scalar?
>
> $Q_{\theta}^\text{new}$ is a scalar. In our anonymous codebase (link in appendix lines 547-8), line 36 of `policies_cont/networks/gaussian_critic.py` shows this.
>
> > TD3 uses double Q-learning, which is orthogonal to what's proposed here
>
> We do not think DoubleGum is orthogonal to TD3 as both algorithms induce pessimism. The pessimism of DoubleGum is adjusted by changing the pessimism factor $c$. TD3 computes a pessimistic estimate of the bootstrapped target by taking a sample-wise minimum from the ensemble. Pessimism of DDPG can be varied by using Twin Critics/not.
>
> > it'd be nice to showcase improvements over TD3 too
>
> Our implementations of DDPG with Twin Networks and TD3 are equivalent, and we have thus changed  "DDPG (best w/wo Twin Networks)" to "Best of DDPG/TD3" in our 1-page `.pdf` whenever necessary. Thus, in aggregate, DoubleGum with default hyperparameters outperforms all benchmark algorithms, including TD3 (Figure 14). Also, the pessimism factor hyperparameter $c$ may be adjusted to outperform TD3 both in aggregate across suites (Figure 15) and per suite (Updated Figure 5). More details are presented in Point 1 of our global rebuttal.
>
> > it seems that the performance is sensitive to the hyper-parameter $c$
>
> We intend $c$ in DoubleGum to be similar to how the learning rate is used in stochastic gradient descent optimizers, so sensitivity is preferred. We cover this in Point 4 of our global rebuttal.
>
> > How is $c$ selected? Is $c=0$ the best-performing hyper-parameter in general?
>
> In our attached 1-page `.pdf`, we have created Figure 16, which is Figure 4 aggregated into one graph. We have now changed the default pessimism factor to $c = -0.1$, as it is the value with best aggregate performance.
>
> > lack a solid theoretical grounding as to whether the newly proposed algorithm would converge or not (not formally stated or proved in the paper)
>
> To the best of our knowledge, there are two types of convergence analysis in Q-Learning: 1) operator-theoretic analysis over tabular Q-functions, and 2) training dynamics of neural network parameters. We believe the second is more appropriate for DoubleGum, because our theory addresses issues in using neural networks (and not tables) for Q-learning. DoubleGum uses variance networks, but we are unaware of any literature analyzing their training dynamics. We thus believe convergence guarantees are beyond the scope of this rebuttal (and our paper)
>
> Nevertheless, we are happy to provide an intuition about convergence we are happy to include in our paper. First, consider the following a numerically equivalent loss function to that of DoubleGum in Equation 27: $\sigma_{\bar{\theta}} \log \sigma_\theta + \frac{\epsilon_\theta^2}{\sigma_\theta}$. For each value of $\epsilon_\theta^2$, our resultant function is convex wrt the numerical value of $\sigma_\theta$. If $\sigma_\theta$ is large, the loss is dominated by $\sigma_{\bar{\theta}} \log \sigma_\theta$ and gradient descent will minimize $\sigma_\theta$. If $\sigma_\theta$ is small, the loss is dominated by $\epsilon_\theta^2$, and we can now rely on convergence guarantees of l2 Q-learning.
>
> The guarantee we use is DR3 [2], which states convergence if $\phi_\theta(s, a)^\top \phi_\theta(s, a) - \gamma \phi_\theta(s, a)^\top \phi_\theta(s, a) \geq 0$, where $\phi(s, a)$ is the penultimate layer of the network. (This was also found by [3]). Following work [4] satisfied this condition by normalizing the layer before the value head. To do this, we used GroupNorm with 16 groups as it had empirically stronger performance. In 100 training runs on three of the hardest environments (humanoid-run, Humanoid-v4, and BipedalWalkerHardcore-v3), all seeds resulted in positive learning curves. We would be happy to include this ablation in our appendix.
>
> [1] Seitzer, M. et al. On the pitfalls of heteroscedastic uncertainty estimation with probabilistic neural networks.
>
> [2] Aviral Kumar et al. DR3: Value-Based Deep Reinforcement Learning Requires Explicit Regularization.
>
> [3] Wang, Z. T. and Ueda, M. A convergent and efficient deep Q network algorithm.
>
> [4] Kumar, A. et al. Offline q-learning on diverse multi-task data both scales and generalizes.
>
> To reiterate, for the reasons outlined above, we believe that theoretical guarantees for the convergence of DoubleGum are outside the scope of our work.
> Nevertheless, we hope the additional results we have presented showcase the promise of our new approach.

---

> > ### Comment · Reviewer_uPm8 · 2023-08-20
> > **Thanks for the reply**
> >
> > Thank you for the reply.
> >
> > == **$Q_\theta(s)$ is a random variable because it varies throughout training.** ==
> >
> > The fact that $Q_\theta$ varies during training due to changes in $\theta$ does not mean that we model it as a random variable. For any fixed training run, the sequence of $\theta$ is deterministic. Aggregating over all possible training runs, and accounting for randomness in the training process, we can say $Q_\theta$ has a marginal distribution at any fixed iteration during training. It is not clear to me whether the aim here is to model such a distribution, and how fundamentally one can do this, since we only have one training run in practice.
> >
> > == **In supervised learning, it is standard to model the output of a neural network as a random variable, in regression the neural network estimate is a random variable that differs from the optimal value by Normal noise.**==
> >
> > Do you have references for this statement? It is not clear to me why it is fundamentally possible to model the prediction $Q_\theta(s)$ as being just a Guassian variable away from the optimal Q-value. What is the cause of this randomness? According to author's response, it seems that this randomness is due to the training process? Can we always guarantee that $Q_\theta(s)-Q^*$ is Gaussian distributed and hence zero-mean? It should be fairly easy to construct examples where the difference $Q_\theta(s)-Q^*$ is mostly negative and cannot be Gaussian distributed.
> >
> > === **The guarantee we use is DR3 [2], which states convergence if ...** ===
> >
> > I am not super familiar with the reference DR3 [2] but the condition $\phi(s,a)^T\phi(s,a)-\gamma\phi(s,a)^T\phi(s,a)\geq 0$ is trivially satisfied as long as $\gamma\in[0,1]$, no? Since $\phi(s,a)^T\phi(s,a)$ is a non-negative scalar. I might be missing something obvious here or the statement is slightly off?
> >
> > === **To the best of our knowledge, there are two types of convergence analysis in Q-Learning: 1) operator-theoretic analysis over tabular Q-functions, and 2) training dynamics of neural network parameters** ===
> >
> > Case (2) should provide an even stronger and more general statement than case (1), and often time the theoretical guarantee of case (2) is obtained via the properties of the contractive opertor (e.g., linear TD). The algorithm proposed by the authors seems to be agnostic to the parameterization $Q_\theta$, and in the case when the parameterization is tabular, we should arguably be able to obtain tabular convergence to $Q^*$. This would be a valuable result to have.
> >
> > Overall, I think it would be necessary to rigorously state the theoretical convergence of the algorithm as a major part of the paper. Otherwise my overall impression is that maybe the paper has proposed an interesting algorithm with some improvements (with empirical caveats), but it is not clear if this algorithm is principled enough and has solid guarantee.
> >
> > Since the various assumptions made in the paper (e.g., $Q_\theta - Q^*$ is Gaussian distributed) are not necessarily grounded, I'd love to see more theoretical properties entailed by the algorithm. At the end of the day, if the algorithm is provably convergent, we can still see it as a principled improvement over baselines. However, in the current stage, it feels like the paper still falls short in this aspect. The paper does provide empirical gains in certain cases, but to me it would be more satisfying to see such improvements are not just "tricks", but rather principled improvements.
> >
> > I'd like to keep my score as a result.

---

> > > ### Author Response · Authors · 2023-08-20
> > > **Convergence proof in the tabular setting; a more rigorous convergence discussion in the deep learning setting**
> > >
> > > Many thanks for your detailed response!
> > >
> > > > $Q_\theta$ is a random variable
> > >
> > > $Q_\theta$ is defined as a random variable in Equation 9. Our motivation for the form of Equation 9 comes from the equation at the top of Page 3 of Thrun and Schwartz, 1993, *Issues in Using Function Approximation for Reinforcement Learning*. This equation treats $Q_\theta$, the $Q$-function with function approximation as a random variable produced by $Q_\theta(s, a) = Q^\star(s, a) + y_{s, a}$ (notation edited to match ours for clarity), where $y_{s, a}$ is a noise source whose distribution the authors do not yet specify, but we argue is heteroscedastic Gumbel (Equation 9). The authors say that the noise is introduced by the function approximator (bottom of Page 2).
> > >
> > > > For any fixed training run, the sequence of $\theta$ is deterministic.
> > >
> > > This is not true because we update $\theta$ by stochastic gradient descent which involves (probabilistically) sampling a minibatch from the replay buffer.
> > >
> > > > Do you have references for this statement?
> > >
> > > Section 3.1.1 *Maximum Likelihood and Least Squares* from Bishop, Pattern Recognition And Machine Learning, Springer 2006 has Equation 3.8: $p(t \mid x, \mathbf{w}, \beta) = N(t \mid y(x, \mathbf{w}), \beta^{-1})$, which may also be found in Section 5.5.1 *Conditional Log-Likelihood and Mean Squared Error* from Goodfellow et al., Deep Learning, MIT Press 2016, and Equation 11.1 from the draft `.pdf` of Murphy, *Probabilistic Machine Learning: An Introduction* MIT Press, March 2022. This equation is equivalent to $t = \epsilon + y(x, \mathbf{w}), \epsilon \sim N(\epsilon \mid 0, \beta^{-1})$. If $y(x, \mathbf{w})$ is a neural network with parameters $\mathbf{w}$ and the optimal value is produced from x by an unknown underlying optimal function, then the neural network estimate differs from the optimal value by Normal noise.
> > >
> > > However, we realise that as our work concerns RL, drawing an analogy to supervised learning may be confusing, and if accepted, we will remove Appendix A.1 and instead use Thrun and Schwartz, 1993 as precedence for modelling the function approximator as a random variable differing from the optimal model by another random variable.
> > >
> > > > It is not clear to me why it is fundamentally possible to model the prediction $Q_\theta$ as being just a Guassian variable away from the optimal Q-value.
> > >
> > > We would like to point out that we assume that $Q_\theta$ and $Q^\star$ differ by a *Gumbel*, not a *Normal/Gaussian* variable away (Equation 9 in our paper). To quote your original review, we show that "regular L-2 loss based Q-learning can be understood as maximum-likelihood estimation with Gaussian noise" and we share your concern that using a Normal/Gaussian is problematic, and indeed show histograms (Updated Figure 1 in the 1-page `.pdf`) that this assumption is problematic.
> > >
> > > > the DR3 statement is slightly off?
> > >
> > > It should be $\phi(s, a)^\top \phi(s, a) - \gamma \phi(s, a)^\top \phi(s', a') \geqslant 0$, with $s'$ and $a'$ occurring at the next timestep from $s$ and $a$.
> > >
> > > We recently found [1], which provides PAC-bounds backing up our convergence intuition. [1] states that a heteroscedastic regression loss (ours is based on this -- line 167) converges if $Q$ is near $y^\star$ (Section 4, Paragraph 1). Note that our use of DR3 has $Q$ converging to $y^\star$.
> > >
> > > [1] Zhang et al., 2023, Risk Bounds on Aleatoric Uncertainty Recovery
> > >
> > > > and in the case when the parameterization is tabular
> > >
> > > DoubleGum learns $Q_\theta$ and $\sigma_\theta$, so in the tabular setting, two tables would need to be learned, $Q(s, a)$ and $\sigma(s)$. Like SAC, DoubleGum would follow the operation of soft policy iteration and its convergence would have a similar proof. The policy improvement step remains the same as SAC (as discussed in Lines 217 - 221, Appendices A.3, A.4) and would not need to use table $\sigma$. The policy evaluation step would update the two tables $Q$ and $\sigma$.
> > >
> > > $\sigma(s) \leftarrow Variance_a[y^\star(s, a)]$ and $Q(s, a) \leftarrow y^\star(s, a)$, where the target $y^\star$ is defined in Equation 22. Equation 22 may be rewritten with an augmented reward $r_{\text{aug}} = r + \gamma \beta(s') c$, where $\beta(s) = \frac{\sqrt{3}}{\pi} \sigma(s)$ (rearranged from Equation 27) so convergence may guaranteed by a similar argument to Lemma 1 of [2], regardless of how $\sigma$ is updated, just as SAC in the tabular case is guaranteed to converge regardless of how its temperature parameter changes. Note that in DoubleGum, $\beta(s')$ may be thought of as a state-dependent temperature parameter, as previously mentioned in Lines 218-8 and Appendix A.4.
> > >
> > > [2] Haarnoja et al., 2018, Soft Actor-Critic: Off-Policy Maximum Entropy Deep Reinforcement Learning with a Stochastic Actor
> > >
> > > We would be happy to include both the above discussions about convergence in our work.
> > >
> > > We would also be very grateful if you could quickly respond to these points, as there is not much time left for discussion!

---

> > > > ### Author Response · Authors · 2023-08-20
> > > > **DR3: A Brief Derivation**
> > > >
> > > > > I am not super familiar with the reference DR3
> > > >
> > > > Using standard RL notation, the Mean-Squared Bellman Error over one sample is:
> > > > \begin{align*}
> > > >     L_\theta = \frac{1}{2}\epsilon_\theta^2 =
> > > >         \frac{1}{2}
> > > >         \left(
> > > >             Q_\theta(s, a)
> > > >             - r - \gamma \max_{a'} Q_{\bar{\theta}}(s', a')
> > > >         \right)^2
> > > > \end{align*}
> > > > The semi-gradient update rule with learning rate $\eta$ is:
> > > > \begin{align*}
> > > >     \theta \leftarrow \theta - \eta \epsilon_{\theta} \nabla_\theta Q_\theta(s, a)
> > > > \end{align*}
> > > > Note that $\delta_\theta = - \eta \epsilon_{\theta} \nabla_\theta Q_\theta(s, a)$. We would like to see how this change in parameters affect the loss function $L_{\theta + \delta\theta}$. This is achieved by taking a first-order Taylor expansion of $L_{\theta + \delta\theta}$ with perturbation $\delta\theta$. To compute this value, we first take a first-order Taylor expansion of the online $Q$:
> > > > \begin{align*}
> > > >     Q_{\theta + \delta\theta}
> > > >     &= Q_{\theta} + \delta\theta^\top \nabla_\theta Q_{\theta} + \cdots
> > > > \end{align*}
> > > > and the target $Q$:
> > > > \begin{align*}
> > > >     Q^\prime_{\theta + \delta\theta}
> > > >     = Q^\prime_{\theta + \delta\theta}
> > > >     = Q^\prime_{\theta} + \delta\theta^\top \nabla_\theta Q^\prime_{\theta} + \cdots
> > > > \end{align*}
> > > > where $Q_\theta(s, a)$ and $\max_a Q_\theta(s', a)$ are respectively abbreviated as $Q_{\theta}$ and $Q^\prime_{\theta}$. These yield a Taylor expansion of $L_{\theta + \delta\theta}$:
> > > > \begin{align*}
> > > >     L_{\theta + \delta\theta}
> > > >     &= \frac{1}{2} (Q_{\theta} + \delta\theta^\top \nabla_\theta Q_{\theta} - r - \gamma (Q^\prime_{\theta} + \delta\theta^\top \nabla_\theta Q^\prime_{\theta}) )^2
> > > > \end{align*}
> > > > \begin{align*}
> > > >     &= \frac{1}{2} (\epsilon_\theta + \delta\theta^\top (\nabla_\theta Q_{\theta} - \gamma \nabla_\theta Q^\prime{\theta}) )^2
> > > > \end{align*}
> > > > \begin{align*}
> > > >     &= L_\theta + \epsilon_\theta \delta\theta^\top (\nabla_\theta Q_{\theta} - \gamma \nabla_\theta Q^\prime_{\theta}) + \cdots
> > > > \end{align*}
> > > > and thus
> > > > \begin{align*}
> > > >     \delta L
> > > >     = L_{\theta + \delta\theta} - L_{\theta}
> > > >     = \epsilon_\theta \delta\theta^\top (\nabla_\theta Q_{\theta} - \gamma \nabla_\theta Q^\prime_{\theta})
> > > > \end{align*}
> > > > Substituting $\delta\theta$ yields:
> > > > \begin{align*}
> > > >     \delta L
> > > >     &= - \eta \epsilon_\theta^2 \nabla_\theta Q_{\theta}^\top (\nabla_\theta Q_{\theta} - \gamma \nabla_\theta Q^\prime_{\theta})
> > > > \end{align*}
> > > > The Q-Learning loss is a squared loss and bounded below at 0. As the loss is minimized iteratively by gradient descent, the convergence of the loss may be shown by a negative change in the loss at every iteration. This convergence is only guaranteed if $\delta L < 0$ which implies:
> > > > \begin{align*}
> > > >     \nabla_\theta Q_\theta^\top (\nabla_\theta Q_\theta - \gamma \nabla_\theta Q^\prime_\theta) \geqslant 0
> > > > \end{align*}
> > > > Now, with linear function approximation, $Q_\theta(s, a) = \theta^\top \phi(s, a)$, and $\nabla_\theta \theta^\top \phi(s, a) = \phi(s, a)$, so we recover
> > > > \begin{align}
> > > >     \phi(s, a)^\top \phi(s, a) - \gamma \phi(s, a)^\top \phi(s', a') \geqslant 0
> > > > \end{align}
> > > > as desired.

---

### Official Review · Reviewer_WFvy · 2023-07-06

**Soundness:** 2 fair
**Presentation:** 3 good
**Contribution:** 2 fair
**Rating:** 6
**Confidence:** 3

**Summary:**

Typically, TD learning assumes that the TD error follows a normal distribution with a fixed variance (induced by the Bellman squared loss). The authors argue that this assumption is too coarse in practice since the maximization of noisy Q-values across actions (when backing up the value in TD learning) is usually not Gaussian. To fix this, the authors discuss the proper limiting distribution of the Q-function when accounting for the Q-function maximization, and provides a practical temporal-difference backup algorithm that can accurately capture the distribution in the discrete control case and approximately capture it in the continuous control case. On a range of simulated robotic and control tasks, the proposed method is able to achieve better performance compared with existing approaches that naively backup the noisy target (without considering the interaction between the noise and the action maximization).

**Strengths:**

To the best of my knowledge, this is the first paper that uses Gumbel distributions to model TD errors. The main strength of the paper is in the theoretical analysis around this TD error model and a theoretically justified, novel algorithm that can perform approximated TD backup with this new TD error model. The writing is clear and very easy to follow.

**Weaknesses:**

The main weakness of the paper is the empirical evaluation. Overall, I am not convinced that 1) assuming that TD errors follow a homoscedastic normal distribution is problematic on the domains of tasks being tested, 2) the proposed method is able to improve upon existing approaches on these tasks.


- Section 4: I am not sure I am convinced that the Logistic model fits the empirical data better than the Gaussian model (from Figure 1). It actually makes me wonder how much is actually needed in practice to accurately account for the Gumbel noise. When do you expect the gap between these two models to be bigger?
- L196-197: How is the error bar computed? It seems a bit problematic to claim that the normal distribution is a suitable approximation given that their error bars overlap since it could be just due to the distribution variance.
- Just from Figure 3 it is hard to tell whether DoubleGum is better than MoG-DDPG. DoubleGum is slightly better on MuJoCo but slightly worse on both DMC and Box2D.

**Questions:**

- L170: $\sigma(s)$ => $\sigma(s, a)$?
- I wonder how much of the problem in the assumption (that TD error is assumed to be Gaussian) can already be addressed by using distributional RL.
- It is very interesting to see that the method has a parameter that can adjust for the degree of pessimism more smoothly than the twin networks in TD3. There are also methods like RedQ that can adjust this with more fine-grain controls which has not been considered by the authors (though admittedly RedQ would be more expensive as it requires more critic ensemble elements).

**Limitations:**

The authors have adequately addressed the limitations.

---

> ### Author Rebuttal · Authors · 2023-08-09
>
> Many thanks for your review!
>
> We are very happy you highlighted our strong theoretical analysis of TD-errors, our justification for our resultant algorithm (DoubleGum), and that you found the writing clear and very easy to follow.
>
> > [not convinced that] the proposed method is able to improve upon existing approaches
>
> We have uploaded Figures 14 and 15 in our attached 1-page `.pdf`, which show aggregate learning curves across all 33 continuous control tasks. In these graphs, DoubleGum outperforms all benchmark algorithms. More details are given in point 1 of our global rebuttal.
>
> > not sure I am convinced that the Logistic model fits the empirical data better than the Gaussian model (from Figure 1)
>
> We expect little difference in fit between the *hetero*scedastic Normal and *hetero*scedastic Logistic. However, we also expect a vast difference in fit between a *hetero*scedastic Logistic and the *homo*scedastic Normal. We have updated Figure 1 to show this, and it is presented most prominently in Figure 1c. Updated Figure 1a fits homoscedastic distributions and Updated Figure 1b fits heteroscedastic distributions. Visually, heteroscedastic distributions fit far better than homoscedastic ones. More details are in point 3 of our global rebuttal.
>
> > How much is actually needed in practice to accurately account for the Gumbel noise
>
> Due to Updated Figure 1c, we believe we can do this with the heteroscedastic Normal.
>
> > When do you expect the gap between these two models to be bigger?
>
> In Updated Figure 1c, the gap between the homoscedastic Normal and the heteroscedastic distributions increases during training. However, the gap between both heteroscedastic distributions remains close throughout.
>
> > L196-197: How is the error bar computed?
>
> As mentioned in lines 185-6, "The line and error bars in Figure 1c reflect the empirical mean and standard deviation of the fitted NLLs computed over 12 training runs". The same procedure is used in Updated Figure 1.
>
> > Is seems a bit problematic to claim that the normal distribution is a suitable approximation given that their error bars overlap?
>
> Due to Updated Figure 1c, we claim that the heteroscedastic Normal is a suitable approximation, but the homoscedastic Normal is not.
>
> > Just from Figure 3 it is hard to tell whether DoubleGum is better than MoG-DDPG.
>
> Figure 14 in our 1-page `.pdf` is an aggregate version of Figure 3. This new graph shows that in aggregate, DoubleGum outperforms MoG-DDPG. More details are presented in Point 1 of our global rebuttal.
>
> > L170: $\sigma(s)$ => $\sigma(s, a)$?
>
> This is indeed a typo!
>
> > how much of the problem in the assumption (that TD error is assumed to be Gaussian) can already be addressed by using distributional RL
>
> We added a new baseline: QR-DDPG, which combines Quantile Regression [1] with DDPG. Following the original implementation, we learn 201 discrete quantiles. Pessimism of QR-DDPG is adjusted by deciding whether to use Twin Critics/not. Twin-QR-DDPG trains an ensemble of two Quantile Critics, and selects the minimum of two quantile estimates to compute the bootstrapped estimates. Results are reported in our 1-page `.pdf`.
>
> DoubleGum outperforms QR-DDPG with default hyperparameters (Figure 14) and tuned pessimism per-suite (Figure 15). With default hyperparameters, DoubleGum outperforms QR-DDPG, with a bit of overlap in DeepMind Control and MetaWorld (Updated Figure 3), and comfortably outperforms QR-DDPG with tuned pessimism per suite (Updated Figure 5). In aggregate, MoG-DDPG outperformed QR-DDPG (Figure 14). Note that MoG-DDPG is a form of distributional RL and as mentioned in lines 236-7, DoubleGum may be considered a special case of MoG-DDPG.
>
> Our theory showed that TD-errors follow a heteroscedastic Logistic, and we believe that modeling this distribution should be sufficient for distributional RL. We hypothesize that more complex distributions considered by QR and MoG might overfit, but this is a question beyond the scope of our work.
>
> Finally, there are empirically larger gains in performance by adjusting pessimism (ie choosing DDPG or TD3) vs choosing to use distributional RL/not (QR-DDPG vs DDPG) (Figure 13, attached `.pdf`). While it is important to model the distribution of TD-errors, another important component is adjusting pessimism, and DoubleGum presents a computationally efficient way of doing both.
>
> [1] Dabney, W., Rowland, M., Bellemare, M. G., and Munos, R. Distributional reinforcement learning with quantile regression.
>
> > methods like RedQ that can adjust this [pessimism] with more fine-grain controls which has not been considered by the authors
>
> As REDQ involves an increased replay ratio, we did not have the time during the rebuttal period to benchmark it. Instead, we introduce a new algorithm which we name FinerTD3. FinerTD3 trains an ensemble of critics and selects the $i$th smallest value within the ensemble as the bootstrapped target per sample. Note TD3 is a special case with an ensemble of 2 and selecting the smallest value. We used an ensemble size of 5. Over all continuous control tasks, we found that it was best to select the second-smallest value in the ensemble. These values were used in Updated Figure 3 and Figure 14 (graphs with default hyperparameters). Per each suite in DeepMind Control, MuJoCo, MetaWorld and Box2D, the best smallest values to select were respectively 5 (largest), 2 (second smallest), 4 (fourth smallest), and 2(second smallest). These values were used in Updated Figure 4 and Figure 15 (graphs where pessimism is tuned per suite).
>
> In aggregate (Figures 14 and 15), FinerTD3 performs marginally poorer than DoubleGum. We believe this is because FinerTD3 adjusts pessimism finer than other baselines, but still not as fine as the continuous scalar in DoubleGum.
>
> We hope that our response better motivates the problem of modeling TD-errors as homoscedastic normal distribution and better showcases the empirical improvements of DoubleGum.

---

### Official Review · Reviewer_dXQf · 2023-07-26

**Soundness:** 3 good
**Presentation:** 3 good
**Contribution:** 2 fair
**Rating:** 6
**Confidence:** 4

**Summary:**

Instead of modeling the TD error with a homoscedastic normal distribution, this paper tries to utilize two heteroscedastic Gumbel distributions for more complex and accurate error modeling. Based on this assumption, the authors presents a modified Q-learning algorithm, DoubleGum for solving discrete and continuous control tasks. In particular, DoubleGum can also achieve the effect of pessimism so as to avoid the overestimation. Empirical results show more stable training and competitive performance across classic discrete and continuous tasks.

**Strengths:**

It's a novel perspective to model the TD-error following the Gumbel distribution.

**Weaknesses:**

1. Double error modeling in section 3.1 seems a little redundant and not intuitive enough；
2. Though linking the DoubleGum for continuous control to the pessimistic value estimation, the empirical evidence is lacking.
3. Experimental results don't show outstanding performance improvement across multiple benchmarks, and lack of comparison with similar algorithms, e.g. extreme Q-learning[1]. In addition, It is shown that the Gumbel error model helps reduce the overestimation issue, which is quite natural considering its close connection with SAC. So why not compare its behavior with SAC in more detail?

[1] Garg, Divyansh, et al. "Extreme Q-Learning: MaxEnt RL without Entropy." The Eleventh International Conference on Learning Representations. 2023.


**Questions:**

1. What's the motivation of using an additional Gumbel noise for the new function approximator $Q^{new}$ in Sec.3.1? Because this modeling does not seem to involve the Extreme Value Theorem, why to model the error using the same Gumbel distribution?

2. About the DoubleGum for the continuous control, it's said that, for ease of implementation, the pessimism factor $c$ is setted as 0, if this, it doesn't look much different than the common off-policy methods;

3. Sec.3.2 tries to connect the DoubleGum with the pessimistic value estimation, but the experimental results only contain the performance comparison for different choices of $c$, It is better to compare the magnitude of the value function directly;

4. Motiviated by a similar idea, extreme Q-learning also model the TD-error as Gumbel distribution, so it is better to make a detailed comparison between the both algorithms, including methodological differences and experimental performance.

5. About Fig.1a and Fig.1b, there is little difference for fittings by Logistic distribution or Normal distribution. I am not sure that it is a good evidence to justify the proposed error model.

---

> ### Author Rebuttal · Authors · 2023-08-09
>
> Many thanks for your review!
>
> We are very happy that you found our modeling of the TD-error novel.
> > Experimental results don't show outstanding performance improvements across multiple benchmarks
>
> We have uploaded Figures 14 and 15 in our attached 1-page `.pdf`, which show aggregate learning curves across all 33 continuous control tasks. In these graphs, DoubleGum outperforms all benchmark algorithms. More details are given in point 1 of our global rebuttal.
>
> > Lack of comparison with similar algorithms [extreme Q-learning (XQL), SAC].
>
> We have added the SAC and XQL benchmarks in Updated Figures 3 and 5 as well as Figures 14 and 15 in In our attached 1-page `.pdf`. In brief, DoubleGum outperforms SAC and XQL in aggregate. More details of how SAC was benchmarked are given in point 2 of our global rebuttal.
>
> Our XQL benchmark was based on TD3 as it marginally outperformed SAC (Last line of Appendix D.6). Pessimism of XQL was varied by the use of Twin Networks/not, respectively yielding Twin-XQL and XQL (equivalent to what the authors name X-TD3 and X-TD3-DQ). The value of $\beta$ in XQL varies per task, but we vary them per suite, to be consistent with our other algorithms. We sweep over $\beta$s of 3, 4, 10, 20 for Twin-XQL and 1, 2, 5 for XQL, consistent with Appendix D.6 in XQL. Over all continuous control tasks, the best $\beta$ values were 20 and 5 for Twin-XQL and XQL respectively. These values were used in Updated Figure 3 and Figure 14 (graphs with default hyperparameters). For each suite in DeepMind Control, MuJoCo, MetaWorld, and Box2D, the best $\beta$ values were respectively 3, 20, 20, and 4 for Twin-XQL and 1, 5, 5, and 5 for XQL. These values were used in Updated Figure 4 and Figure 15 (graphs where pessimism is tuned per suite).
>
> DoubleGum outperforms XQL with both default hyperparameters (Figure 14) and tuned pessimism per suite (Figure 15). Over each suite, DoubleGum outperforms XQL in all suites apart from Box2D, both with default hyperparameters (Updated Figure 3) and tuned pessimism per suite (Updated Figure 5).
>
> > What's the motivation of using an additional Gumbel noise for the new function approximator $Q^\text{new}$?
>
> We introduced $Q_\theta^\text{new}$ to write a Bellman Equation with one heteroscedastic Logistic noise source (Equation 19), instead of a Bellman Equation written with $Q_\theta$ with two heteroscedastic Gumbel noise sources (Equation 15). The additional Gumbel noise source shifts $Q_\theta$ into $Q_\theta^\text{new}$ using the property of the log-sum-exp (Appendix B.2). We realize that a simpler way of presenting this is to absorb the Gumbel noise source, simplifying Section 3.1 by removing Equations 16-18.
>
> > it's said that, for ease of implementation, the pessimism factor is setted as 0, if this, it doesn't look much different than the common off-policy methods
>
> We have now changed the default pessimism factor to $c = -0.1$ as this gives us empirically better per-suite performance (Updated Figure 3) and aggregate performance with fixed hyperparameters (Figure 14). We would like to point out that functionally, our algorithm uses variance networks unlike the common off-policy methods, which is not common in Q-Learning.
>
> > Empirical evidence is lacking [for pessimistic value estimation ...] compare the magnitude of the value function directly)
>
> This is a great suggestion, and we have added Figure 17 in our 1-page `.pdf'`. This graph shows how the magnitude of the target value function averaged over 256 training samples varies during training. The line and error bars show the mean $\pm$ standard deviation of the averaged target value function. Four graphs are presented, showing results of one task from each suite. In each graph throughout training, the magnitude of the target value function is smaller for a smaller $c$, although there is some overlap between the error bars.
>
> > Make a detailed comparison between [our algorithm DoubleGum and XQL], including methodological differences and experimental performance
>
> We would be happy to include a comparison of XQL and DoubleGum. We have detailed the experimental comparison above. Methodologically, in brief, XQL models TD-errors using one homoscedastic Gumbel, whereas DoubleGum models the same errors with two heteroscedastic Gumbels. This leads to a different loss function of the critic: XQL uses Gumbel regression with a loss derived from the MLE of a Gumbel, whereas our loss function is derived from moment matching the heteroscedastic Logistic (derived by combining the two heteroscedastic Gumbels) with a heteroscedastic Gaussian. While DoubleGum learns the heteroscedastic spread parameter continually throughout training, the spread parameter in XQL is a hyperparameter defined at the beginning of training and fixed throughout.
>
> > About Fig.1a and Fig.1b, there is little difference for fittings by Logistic distribution or Normal distribution.
>
> We expect to see little difference in fittings between a *hetero*scedastic Normal and the *hetero*scedastic Logistic. However, we also expect to see a vast difference in fit between a *hetero*scedastic Logistic and the *homo*scedastic Normal. We believe that did not make the distinction between hetero and homo Normals is clear enough in Figure 1. Therefore, we have updated Figure 1 in our attached 1-page `.pdf`. More details are in point 3 of our global rebuttal.
>
> Updated Fig.1a fits homoscedastic distributions and Updated Fig.1b fits heteroscedastic distributions. Visually, the heteroscedastic distributions fit the underlying histogram far better than the homoscedastic distributions. Also, Updated Fig.1c matches our expectation and shows little difference between hetero-Logistic and hetero-Normal, but a vast difference with the homo-Normal.
>
> We hope that our response clarifies the motivation for the mathematics in Section 3.1, and presents compelling evidence for DoubleGum's pessimistic value estimation.

---

> > ### Comment · Reviewer_dXQf · 2023-08-18
> >
> > Thanks for your detailed response and I would maintain my orginal score.

---

### Author Rebuttal · Authors · 2023-08-09

Many thanks to all reviewers for their comments and feedback!

We are delighted that all five reviewers mentioned the strong experimental analysis of our DoubleGum algorithm and that four reviewers highlighted the novelty of our theoretical analysis (dXQf, WFvy, uPm8, Mh9L).  In addition, we are also very happy that four reviewers marked the presentation as 3 (dXQf, WFvy, je5E, Mh9L), with two explicitly mentioning the clarity of writing (WFvy, Mh9L).

There were four common issues raised by our reviewers.

**1. Three reviewers (dXQF, WFvy, Mh9L) were concerned that we did not clearly show empirical improvements of our algorithm (DoubleGum) over baselines.**  We have better presented DoubleGum's empirical improvements over all baselines in Figures 14 and 15 in our 1-page `.pdf`.  These graphs aggregate all learning curves over all tasks in all suites into one graph.  They show that in aggregate, DoubleGum outperforms all baselines when all methods use default hyperparameters (Figure 14) and also when all methods' pessimism hyperparameters are tuned per suite (Figure 15).

**2. Three reviewers (dXQf, je5E, and Mh9L) suggested benchmarking the algorithm SAC.**  We have benchmarked SAC and added its aggregate learning curves to Updated Figures 3 and 5 as well as Figures 14 and 15 in our 1-page `.pdf`.  Pessimism of SAC is tuned by varying the use of Twin Networks/not, just as we do with TD3/DDPG in Updated Figure 3, as specified in line 250.  In aggregate, DoubleGum outperforms SAC with both default hyperparameters (Figure 14) and tuned pessimism per suite (Figure 15).  Finally, over each suite, DoubleGum outperforms SAC in all suites, both with default hyperparameters (Updated Figure 3) and tuned pessimism per suite (Updated Figure 5).  If accepted, we will upload individual learning curves from each task in the appendix.

**3. Two reviewers (dXQf, WFvy) did not find Figure 1 convincing.**  Figure 1 aims to present empirical evidence showing that the heteroscedastic Normal is a far better approximation of the heteroscedastic Logistic than a homoscedastic Normal.  We have updated Figure 1 in our 1-page `.pdf` to better present this empirical evidence.  No data has been changed between the original and updated Figure 1s.  Instead, we have changed the labels to show which distributions are hetero and homoscedastic, removed distributions in the graph that we do not discuss in the main paper, and added a homoscedastic Normal NLL curve to subgraph 'c' such that the goodness of fit for all three aforementioned distributions may be compared.

**4. Two reviewers (uPm8, Mh9L) queried the sensitivity and stability of our algorithm with respect to our pessimism hyperparameter $c$.**  We intend $c$ in DoubleGum to be used in a similar way to the learning rate in gradient descent optimizers.  We thus intend DoubleGum to be sensitive to $c$ such that performance may be improved by tuning $c$, but we have also presented a default value of $c=-0.1$ (updated from $c=0$ in our paper) that showcases good aggregate performance across all tasks (Figure 14).  The default value is inspired by optimizers with a default learning rate with good aggregate performance to naively try (eg Adam has a lr of 3e-4).

Individual issues raised by the reviewers include suggesting more algorithms to be benchmarked (XQL, distributional RL, an algorithm with finer pessimism control), empirical evidence for the effectiveness of our pessimism factor hyperparameter, a discussion of theoretical convergence guarantees for our algorithm DoubleGum, and clarifications of our mathematics.  We address all individual issues in individual rebuttals.

---

### Decision · Program_Chairs · 2023-09-21

**Decision:**

Accept (spotlight)

**Comment:**

As the generally positive reviews indicate, this paper presents concepts and results that could have broad community interest. There seems to be a difference in viewpoints that kept one reviewer critical.